# FlexMotion: Lightweight, Physics-Aware, and Controllable Human Motion Generation

## Abstract

Lightweight, controllable, and physically plausible human motion synthesis is crucial for animation, virtual reality, robotics, and human-computer interaction applications. Existing methods often compromise between computational efficiency, physical realism, or spatial controllability. We propose FlexMotion, a novel framework that leverages a computationally lightweight diffusion model operating in the latent space, eliminating the need for physics simulators and enabling fast and efficient training. FlexMotion employs a multimodal pre-trained Transformer encoder-decoder, integrating joint locations, contact forces, joint actuations and muscle activations to ensure the physical plausibility of the generated motions. FlexMotion also introduces a plug-and-play module, which adds spatial controllability over a range of motion parameters (e.g., joint locations, joint actuations, contact forces, and muscle activations). Our framework achieves realistic motion generation with improved efficiency and control, setting a new benchmark for human motion synthesis. We evaluate FlexMotion on extended datasets and demonstrate its superior performance in terms of realism, physical plausibility, and controllability.

## 1 Introduction

Generating controllable and realistic human motion is a critical task with applications in various domains, including animation Zhu et al. (2023), sports and rehabilitation Yang et al. (2023); Zhang et al. (2024a); Cheng et al. (2023); Tashakori et al. (2022), virtual reality Zhu et al. (2023), robotics Tashakori et al. (2024), and human-computer interaction Arkushin et al. (2023); Zhang et al. (2022); Servati et al. (2024). Human motion involves complex interactions between joint movements, contact forces, and muscle activations, necessitating a comprehensive approach that can capture both kinematic and dynamic aspects. Despite the remarkable progress in human motion generation, challenges remain in developing models that effectively balance physical realism, computational efficiency, and fine-grained controllability.

Traditional methods often fail to control the intricate biomechanics of human movement, which involve complex interactions between kinematics, dynamics, and environmental context Tripathi et al. (2023b); Zhang et al. (2024b); Xie et al. (2021a); Chiquier & Vondrick (2023). This deficiency is particularly notable in applications such as sports and rehabilitation, where the precision of muscle activations and contact forces is crucial for accurate simulation Chiquier & Vondrick (2023). Furthermore, current methods focused on physical plausibility often demand high computational resources, such as physics engines, rendering them impractical for real-time applications Yuan et al. (2023); Xie et al. (2021a); Tripathi et al. (2023a).

We propose FlexMotion, a novel, lightweight, and physics-aware framework that generates multimodal human motion sequences conditioned on text and diverse kinematics and dynamics information. FlexMotion leverages a multimodal, physically plausible pre-trained Transformer encoder-decoder, learning the relationship between joint trajectories, contact forces, joint actuations, and muscle activations to ensure that the generated motions are aligned with human biomechanics. FlexMotion operates in the latent space, significantly reducing the computational cost for training and inference compared to traditional human motion generation methods. Our model also introduces a spatial controllability module that allows for fine-grained control over spatial, muscle, joint actua-

Figure 1: **The proposed FlexMotion can generate physically-plausible human motion sequences using text prompt and spatial control over diverse motion kinematic properties,** including (a) contact forces, (b) joint locations, (c) muscle activation, and (d) joint actuation.

tion, and contact force parameters, enhancing the applicability of generated motions across various domains.

FlexMotion's capabilities can be best understood through examples of its generated data. We present a few instances in Fig. 1. In Fig. 1(a), a person performs a handspring, demonstrating FlexMotion's performance by combining contact forces and text. In Fig. 1(b), a person walks on a wavy path, showcasing FlexMotion's spatial controllability over movement trajectory. In Fig. 1(c), a person walks in a straight line, where the output is controlled using quadriceps activation over time. In Fig. 1(d), a person moves their left hand while sitting on the ground, where FlexMotion was given a combination of forearm actuation and textual description. For each of these examples, motion generators conditioned only on text descriptions might generate a wide range of possibilities that may not align with the user's intent. By integrating the spatial conditioning module, we can control the generated motion to follow specific trajectories, contact forces, joint actuations, and muscle activations, enabling precise and contextually appropriate motion generation.

Our main contributions are as follows:

- **Physical Plausibility**: We propose the first method that ensures generated motions are physically plausible by training a Transformer encoder-decoder with physical constraints. We extend current datasets to include muscle activations, contact forces, and joint actuations, enabling multimodal sensor fusion.

- **Computational Efficiency**: Our diffusion model operates in the latent space, making it lightweight and fast to train and infer without requiring complex physics simulators for further correction.

- **Enhanced Controllability**: FlexMotion provides plug-and-play fine-grained control over spatial, muscle, joint activation, and contact force parameters, enhancing the applicability of generated motions across various domains.

The following section discusses related works in human motion generation, physics-aware human motion modeling, and controllability. Next, we describe our proposed method, including mathematical formulations and architectural details. We then discuss our experimental results on popular datasets, including HumanML3D Guo et al. (2022a), KIT-ML Plappert et al. (2016), and FLAG3D Tang et al. (2023). Finally, we conclude with insights and directions for future work.

## 2 RELATED WORK

### 2.1 HUMAN MOTION GENERATION

Motion generation literature has focused on two main approaches: first, using autoregressive models, which use past generated frames to generate the subsequent frames recursively Zhu et al. (2023). Second, sequence-based models which generate the entire sequence at once Zhu et al. (2023); Feng et al. (2024); Lou et al. (2023); Zhong et al. (2023); Xu et al. (2023); Ma et al. (2024); Dabral et al. (2023). In the second approach, researchers have employed a variety of generative models to

achieve this, including Generative Adversarial Networks (GANs) Zhu et al. (2023), Variational Autoencoders (VAEs) Zhu et al. (2023); Jiang et al. (2024), Normalizing Flows (NFs) Zhu et al. (2023), and Diffusion Models (DMs) Zhu et al. (2023); Tevet et al. (2023), as well as a task-specific model known as Motion Graphs (MGs) Zhu et al. (2023). Among these approaches, Diffusion Models, particularly Denoising Diffusion Probabilistic Models (DDPMs), have demonstrated promising and diverse results Tevet et al. (2023); Zhu et al. (2023).

DDPMs have been particularly successful due to their ability to model complex data distributions and generate high-quality samples without mode collapse, a common issue in other generative models like GANs Stypułkowski et al. (2024). These models have found applications beyond motion sequence generation, including image synthesis Ruiz et al. (2023); Li et al. (2024); Wang et al. (2024); He et al. (2022); Zhang et al. (2023c;a); Van Le et al. (2023), video generation Zhou et al. (2024); Zhang et al. (2023a); Wu et al. (2023), and other domains where generating realistic sequences is essential.

Motion synthesis research aims to produce realistic and natural human motion patterns under various conditions, leveraging the flexibility and robustness of models like DDPMs. These conditions can include text Tevet et al. (2023); Zhang et al. (2024c; 2023b); Zhao et al. (2023); Chen et al. (2023); Qian et al. (2023), audio Zhu et al. (2023), action Tevet et al. (2023), music Zhu et al. (2023), images Zhu et al. (2023), 3D scene Zhu et al. (2023), spatial contexts Xie et al. (2023); Karunratanakul et al. (2023), objects Zhu et al. (2023), or a combination of such conditions Ling et al. (2023); Jin et al. (2024). These generated sequences can consist of either key point locations, joint rotation Zhu et al. (2023); Delmas et al. (2023), or mesh parameters extracted from models such as SMPL Loper et al. (2015).

Despite the advancements in generating realistic human motion sequences, current models often perform inadequately when tasked with synthesizing complex dynamic movements that adhere to biomechanical and physical laws. These models produce noticeable physical artifacts, such as unnatural joint rotations, unrealistic muscle dynamics, and incorrect contact points during environmental interactions. This deficiency arises especially while generating long motion sequences, primarily because existing approaches lack an explicit understanding of the relative mapping between muscle activations, joint torques, and contact forces, which are crucial for generating physically plausible motions Zhang et al. (2024c). To address these limitations, we employ a pretrained autoencoder that encodes motion properties in the latent space while preserving essential biomechanical information, with a decoder that integrates physics-based constraints to ensure physically accurate motion reconstruction, as detailed in the method section.

## 2.2 Physics-aware Human Motion Modeling

To generate physically plausible motion sequences, researchers have employed two primary approaches: first, interaction with physics simulators Xie et al. (2021b); Lee et al. (2019); Yuan et al. (2023), and second, the integration of physics-based constraints into the reconstruction loss function Tevet et al. (2023). While physics simulators provide detailed physical interactions, they are computationally expensive and non-differentiable Zhang et al. (2024b); Tripathi et al. (2023b), significantly limiting their utility. The non-differentiability obstructs gradient backpropagation, hindering the effective optimization and refinement of generated motions Zhang et al. (2024b); Tripathi et al. (2023b). Alternatively, integrating physics-based constraints directly into generative models offers a more computationally efficient approach to maintaining physical realism without needing full simulation. In the second category, MDM integrates pose consistency, foot placement, and velocity loss Tevet et al. (2023), while IPMAN-R introduces stability and ground interaction loss to enhance motion realism in dynamic environments Tripathi et al. (2023b). PhysPT integrates contact points, force, and Euler–Lagrange consistency loss to accurately simulate physical interactions Zhang et al. (2024b). Additionally, authors in Xie et al. (2021a) incorporate dynamic constraints, contact points, penetration avoidance, and smooth transition terms to produce realistic motion estimation. However, despite these advancements, both approaches struggle to capture complex dynamic motions thoroughly and often fail to adhere to biomechanical laws, leading to noticeable physical artifacts, especially in long sequence generations Zhang et al. (2024c). Our work addresses these challenges by utilizing a pretrained autoencoder architecture to generate physically plausible outputs, detailed in the method section.

## 2.3 Controllable Motion Generation

Controlling motion generation algorithms is essential for creating realistic and contextually appropriate human motions, especially when these motions must adhere to specific spatial constraints or trajectories. Recent advancements have been inspired by techniques from image generation, such as ControlGAN Li et al. (2019), ControlNet Zhang et al. (2023c), ControlNet+ Li et al. (2024), and the T2I adapter Mou et al. (2024); Cao et al. (2024), which offer nuanced control over generated outputs. OmniControl Xie et al. (2023) introduces flexible spatial control signals into a diffusion-based motion model. It allows precise control over various joints over time and improves motion realism Xie et al. (2023). Guided Motion Diffusion (GMD) Karunratanakul et al. (2023) enhances spatial accuracy by integrating constraints such as predefined trajectories and obstacles, using feature projection and dense guidance to ensure coherence between spatial information and generated motions Karunratanakul et al. (2023).

However, both OmniControl and GMD only control joint trajectories and do not account for crucial kinematic parameters such as joint actuation, ground contact forces, or muscle activation, which limits their ability to generate refined motion sequences that adhere to the physical constraints required for realistic and application-specific human movement synthesis. We propose a spatial controllability module to incorporate these critical kinematic and biomechanical factors, enabling more precise motion generation, as detailed in the method section.

## 3 Proposed Method

The objective of FlexMotion is to generate realistic motion sequences $\{x_t\}_{t=1}^T$ conditioned on a text prompt $c$ and a wide range of spatial conditions $\{c_t\}_{t=1}^T$, where $T$ is desired sequence length. The overall architecture of FlexMotion, illustrated in Fig.2, consists of three main components. First, the **Physics-aware Multimodal Autoencoder** (Sec.3.1) learns to map detailed kinematic and dynamic properties of motion to a latent space that captures the essential features of human motion while enforcing physical plausibility through physics-based constraints during motion sequence reconstruction in the decoder. Second, the **FlexMotion Diffusion Model** (Sec.3.2) generates desired latent embedding sequences conditioned on the text prompt. Third, the **Spatial Controllability Module** (Sec.3.3) provides fine-grained spatial control over critical motion parameters, enabling precise and contextually appropriate motion generation. In the following sections, we discuss each component in detail. Further details and pseudocode for both the training and inference processes are provided in the appendix.

### 3.1 Physics-aware Multimodal Autoencoder

To model complex human motions while enforcing physical plausibility, we use a transformer-based autoencoder architecture capable of handling multiple motion modalities, similar to the architecture introduced in Zhang et al. (2024b). Fig. 3 provides an overview of the proposed multimodal autoencoder. At each time step $t$, the motion data $\mathbf{x}_t$ consists of various components, including joint positions $\mathbf{p}_t \in \mathbb{R}^{J \times 3}$, joint rotations $\mathbf{r}_t \in \mathbb{R}^{J \times 3}$, joint velocities $\dot{\mathbf{r}}_t \in \mathbb{R}^{J \times 3}$, joint accelerations $\ddot{\mathbf{r}}_t \in \mathbb{R}^{J \times 3}$, muscle activations $\mathbf{a}_t \in \mathbb{R}^M$, joint torques $\boldsymbol{\tau}_t \in \mathbb{R}^{J \times 3}$, and contact forces $\boldsymbol{\lambda}_t \in \mathbb{R}^{J \times 3}$, where $J$ and $M$ denotes total number of joints and muscles respectively. These modalities are concatenated into a single feature vector, as described in Eqn. 1, where $D$ represents the dimensionality of the input feature vector, calculated based on the dimensions of each modality.

$$\mathbf{x}_t = [\mathbf{p}_t, \mathbf{r}_t, \dot{\mathbf{r}}_t, \ddot{\mathbf{r}}_t, \mathbf{a}_t, \boldsymbol{\tau}_t, \boldsymbol{\lambda}_t] \in \mathbb{R}^D \tag{1}$$

The encoder $\mathcal{E}(.)$ processes the input sequence $\mathbf{x}_t \in \mathbb{R}^D$ and maps it to a sequence of latent representations $\mathbf{x}_t^e \in \mathbb{R}^d$ (Eqn. 2), where $\theta_\mathcal{E}$ is the encoder parameters, and $d$ is latent space dimension where $d \ll D$.

$$\mathbf{x}_t^e = \mathcal{E}(\mathbf{x}_t; \theta_\mathcal{E}) \tag{2}$$

The decoder $\mathcal{D}(.)$ reconstructs the motion sequence from the latent representations $\mathbf{x}_t^e \in \mathbb{R}^d$ (Eqn. 3), with $\theta_\mathcal{D}$ being the decoder parameters. Here, $\hat{\mathbf{x}}_t \in \mathbb{R}^D$ denote the reconstructed output.

Figure 2: **Overview of the proposed FlexMotion framework**. It consists of first, multimodal autoencoder, which maps motion kinematic and dynamic properties to latent space (Sec. 3.1), second, latent space motion diffusion model, which generates a motion sequence in latent space conditioned on text prompt (Sec. 3.2) and third, spatial controllability module, which adds further control to the generated motion (Sec. 3.3).

$$\hat{\mathbf{x}}_t = \mathcal{D}(\mathbf{x}_t^e; \theta_{\mathcal{D}}) \tag{3}$$

To ensure accurate reconstruction of each modality, we define a total reconstruction loss $\mathcal{L}_{\text{recon}}$ as a weighted sum of modality-specific losses (Eqn. 4). The loss terms include the $l_2$ norm between the ground truth and reconstructed values for joint positions, rotations, velocities, accelerations, muscle activations, joint torques, and $l_1$ norm for contact forces since its mostly sparse vector. The parameters $\alpha_{\text{pos}}, \alpha_{\text{rot}}, \alpha_{\text{vel}}, \alpha_{\text{acc}}, \alpha_{\text{torque}}, \alpha_{\text{force}}$, and $\alpha_{\text{muscle}}$ are weighting factors that balance the importance of each modality.

$$\mathcal{L}_{\text{recon}} = \sum_{t=1}^{T} \left[ \alpha_{\text{pos}} \|\mathbf{p}_t - \hat{\mathbf{p}}_t\|_2^2 + \alpha_{\text{rot}} \|\mathbf{r}_t - \hat{\mathbf{r}}_t\|_2^2 + \alpha_{\text{vel}} \|\dot{\mathbf{r}}_t - \hat{\dot{\mathbf{r}}}_t\|_2^2 + \alpha_{\text{acc}} \|\ddot{\mathbf{r}}_t - \hat{\ddot{\mathbf{r}}}_t\|_2^2 \right.$$
$$\left. + \alpha_{\text{torque}} \|\boldsymbol{\tau}_t - \hat{\boldsymbol{\tau}}_t\|_2^2 + \alpha_{\text{force}} \|\boldsymbol{\lambda}_t - \hat{\boldsymbol{\lambda}}_t\|_1^1 + \alpha_{\text{muscle}} \|\boldsymbol{a}_t - \hat{\boldsymbol{a}}_t\|_2^2 \right] \tag{4}$$

To ensure that the generated motions adhere to the laws of physics, inspired by Zhang et al. (2024b); Lee et al. (2019), we incorporate physics-based constraints derived from body dynamics. Specifically, we enforce the Euler-Lagrange equation (Eqn. 5) to ensure the generated motions are physically plausible.

$$\mathbf{M}(\mathbf{r}_t)\ddot{\mathbf{r}}_t + \mathbf{C}(\mathbf{r}_t, \dot{\mathbf{r}}_t)\dot{\mathbf{r}}_t + \mathbf{G}(\mathbf{r}_t) = \boldsymbol{\tau}_t + \mathbf{J}_C^{\top}(\mathbf{r}_t)\boldsymbol{\lambda}_t \tag{5}$$

In this equation, $\mathbf{M}(\mathbf{r}_t)$ represents the mass matrix, $\mathbf{C}(\mathbf{q}_t, \dot{\mathbf{r}}_t)$ accounts for Coriolis and centrifugal forces, $\mathbf{G}(\mathbf{r}_t)$ is the gravitational force vector, and $\mathbf{J}_C(\mathbf{r}_t)$ refers to the contact Jacobian matrix. More details can be found in the appendix.

We define the physics-based loss $\mathcal{L}_{\text{euler}}$ as the $l_2$ norm between the left-hand side and right-hand side of Eqn. 5 (Eqn. 6). This differentiable loss encourages the reconstructed motion to satisfy the physical equations governing the musculoskeletal system (More details in Zhang et al. (2024b); Lee et al. (2019)).

Figure 3: **Overview of Physics-aware Multimodal Autoencoder.** It maps diverse motion properties into the latent space and reconstructs them while enforcing physics-based loss terms (Sec. 3.1).

$$\mathcal{L}_{\text{euler}} = \sum_{t=1}^{T} \left\| \mathbf{M}(\mathbf{r}_t)\ddot{\mathbf{r}}_t + \mathbf{C}(\mathbf{r}_t, \dot{\mathbf{r}}_t)\dot{\mathbf{r}}_t + \mathbf{G}(\mathbf{r}_t) - \boldsymbol{\tau}_t - \mathbf{J}_C^{\top}(\mathbf{r}_t)\boldsymbol{\lambda}_t \right\|^2 \quad (6)$$

Enforcing muscle coordination is also critical for generating realistic and physically plausible motions. We model muscle coordination by computing muscle activations that produce desired joint accelerations while minimizing excessive activation. Following Lee et al. (2019), the muscle loss function $\mathcal{L}_{\text{muscle}}$ is defined as Eqn. 7.

$$\mathcal{L}_{\text{muscle}} = \sum_{t=1}^{T} \left( \|\ddot{\mathbf{r}}_t - L\mathbf{a}_t\|_2^2 + \beta_{\text{reg}} \|\mathbf{a}_t\|_2^2 \right) \quad (7)$$

The matrix $L \in \mathbb{R}^{(J \times 3) \times M}$ maps muscle activations to joint accelerations, which is derived from musculoskeletal dynamics, and $\beta_{\text{reg}}$ serves as a regularization weight to prevent excessive muscle activations. The network is trained to minimize $\mathcal{L}_{\text{muscle}}$, ensuring that the activations produce the desired accelerations while adhering to physical constraints.

The total loss for training the Transformer encoder-decoder is defined as Eqn. 8, where $\beta_{\text{euler}}$ and $\gamma_{\text{muscle}}$ are weighting factors for the physics-based constraints and muscle loss, respectively.

$$\mathcal{L}_{\text{AE}} = \mathcal{L}_{\text{recon}} + \gamma_{\text{euler}}\mathcal{L}_{\text{euler}} + \gamma_{\text{muscle}}\mathcal{L}_{\text{muscle}} \quad (8)$$

We train the physics-aware Transformer encoder-decoder by minimizing the total loss $\mathcal{L}_{\text{total}}$. This involves updating the encoder parameters $\theta_{\mathcal{E}}$, and decoder parameters $\theta_{\mathcal{D}}$. The training ensures that the reconstructed motions match the input data and satisfy physical laws and muscle dynamics.

## 3.2 LATENT SPACE MOTION DIFFUSION MODEL

To generate diverse and realistic motion sequences, we employ a diffusion model operating in the latent space $X^e = \{\mathbf{x}_t^e\}_{t=1}^{T}$ obtained from the trained Transformer encoder $\mathcal{E}(.)$. Following Tevet et al. (2023), we define a forward and reverse diffusion process. An overview of the model can be found in Fig. 2.

The forward process gradually adds Gaussian noise to the latent variables, where $\beta_n$ is a variance schedule. The noise is sampled from a Gaussian distribution with mean $\sqrt{1-\beta_n}\mathbf{X}_{n-1}^e$ and variance $\beta_n\mathbf{I}$ (Eqn. 9).

$$q(\mathbf{X}_n^e | \mathbf{X}_{n-1}^e) = \mathcal{N}(\mathbf{X}_n^e | \sqrt{1-\beta_n}\mathbf{X}_{n-1}^e, \beta_n\mathbf{I}) \quad (9)$$

The reverse process learns to remove noise step by step, where $\boldsymbol{\mu}_\theta$ is a neural network parameterized by $\theta$ that predicts the noise at each iteration $n$ (Eqn. 10).

$$p_\theta(\mathbf{X}_{n-1}^e | \mathbf{X}_n^e) = \mathcal{N}(\mathbf{X}_{n-1}^e | \boldsymbol{\mu}_\theta(\mathbf{X}_n^e, n, c), \sigma_n^2\mathbf{I}) \quad (10)$$

To train the diffusion model, we freeze the pretrained Transformer encoder-decoder and optimize $\mathcal{L}_{\text{diff}}$, where $\mathbf{X}_{n-1}^e = \boldsymbol{\epsilon}_\theta(\mathbf{X}_n^e, n, c)$ Murphy (2023) (Eqn. 11), and $N$ is number of diffusion steps.

$$\mathcal{L}_{\text{diff}} = \mathbb{E}_{X_0^e \sim q(\mathcal{E}(X_0)|c), n \sim \text{Unif}(0, N-1), \boldsymbol{\epsilon} \sim \mathcal{N}(0,1)} \left[ \| \boldsymbol{\epsilon} - \boldsymbol{\epsilon}_\theta(\mathbf{X}_n^e, n, c) \|_2^2 \right] \tag{11}$$

### 3.3 SPATIAL CONTROLLABILITY MODULE

We integrate a spatial controllability module inspired by Zhang et al. (2023c) to enable fine-grained control over the generated motion. At each time step $t$, the control inputs $\mathbf{c}_t \in \mathbb{R}^D$ can include desired joint positions, velocities, muscle activations, or other motion parameters that resemble the modalities in $x_t$. We use the frozen encoder $\mathcal{E}(.)$, trained in section 3.1, to map the sequence of control signals $\{\mathbf{c_t}\}_{t=1}^T$ into a latent space $C^e \in \mathbb{R}^d$. We freeze the trained diffusion model described in section 3.2 and introduce a trainable copy between two zero-initialized convolution layers, $\mathbf{Z}$, as shown in Eqn. 12.

$$\boldsymbol{\epsilon}_{\theta_{\text{total}}}(\mathbf{X}_n^e, C^e, n, c) = \boldsymbol{\epsilon}_\theta(\mathbf{X}_n^e, n, c) + \mathbf{Z}(\boldsymbol{\epsilon}_{\theta_C}(\mathbf{X}_n^e + \mathbf{Z}(C^e, \theta_{z1}), n, c), \theta_{z2}) \tag{12}$$

This design ensures that initially, the control module's output does not interfere with the pretrained diffusion model, allowing the network to learn how to gradually incorporate control conditions during training. The overall learning objective can be described as Eqn. 13.

$$\mathcal{L}_{\text{total}} = \mathbb{E}_{X_0^e \sim q(\mathcal{E}(X_0)|c), C^e, n \sim \text{Unif}(0, N-1), \boldsymbol{\epsilon} \sim \mathcal{N}(0,1)} \left[ \| \boldsymbol{\epsilon} - \boldsymbol{\epsilon}_{\theta_{\text{total}}}(\mathbf{X}_n^e, C^e, n, c) \|_2^2 \right] \tag{13}$$

## 4 EXPERIMENTAL RESULTS

**Datasets.** We evaluate FlexMotion on three popular datasets: **HumanML3D** Guo et al. (2022a), **KIT-ML** Plappert et al. (2016), and **Flag3D** Tang et al. (2023). HumanML3D, derived from AMASS and HumanAct12, contains 14,616 motion sequences with 44,970 textual annotations. KIT-ML provides 3,911 motion sequences with 6,278 descriptions, while Flag3D offers 180,000 videos spanning 60 fitness activities.

**Data Augmentation.** To support physics-aware motion modeling, we augmented these datasets using OpenSim Delp et al. (2007); Seth et al. (2018), a popular and widely acceptable software for biomechanics research and motor control science Delp et al. (2007); Seth et al. (2018), incorporating detailed muscle activations, contact forces, joint positions, rotations, actuation, and velocity. We employed a full-body OpenSim model Van Horn & Team (2016) with 21 body segments, 29 degrees of freedom, and 324 musculotendon actuators, providing rich detail for lumbar movements and trunk muscle dynamics. This augmentation enhances the biomechanical fidelity of the motion data, which is critical for realistic motion synthesis. We provide more details in the appendix.

**Evaluation Metrics.** FlexMotion's performance is comprehensively evaluated across several key metrics, encompassing naturalness, textual relevance, diversity, physical plausibility, and spatial control accuracy. Naturalness is quantified using the **Fréchet Inception Distance (FID)** Tevet et al. (2023), which compares the distribution of generated motions to real data. Textual relevance is measured via **R-Precision** Tevet et al. (2023), assessing how well the generated motions align with textual descriptions. To ensure variability, **diversity (DIV)** is evaluated by computing the pairwise distance between generated motions Tevet et al. (2023). Physical plausibility is verified through metrics like **Foot Skating, Penetration, Contact Force Accuracy, and Joint Actuation Consistency** Xie et al. (2023). Biomechanical plausibility is ensured by checking that **Muscle Activation Limits** stay within realistic physiological constraints Lee et al. (2019). Finally, spatial control accuracy is assessed using the **Trajectory Error metric** Xie et al. (2023), focusing on how well the generated motions adhere to intended spatial trajectories. All results are reported as mean across ten independent runs, ensuring robustness and reproducibility. More details can be found in the appendix.

**Implementation Details.** FlexMotion is built on the MDM framework Tevet et al. (2023), leveraging CLIP Radford et al. (2021) for text encoding and employing classifier-free guidance during

Table 1: **HumanML3D** test set Guo et al. (2022b): Performance comparisons of text-to-motion synthesis methods. The complete table can be found in the appendix.

| Method | R-Precision ↑ | FID ↓ | DIV → | Skate ↓ | Float ↓ | Penetrate ↓ | Contact Force ↓ | Joint Actuation ↓ | Muscle Limit ↓ | Trajectory ↓ |
|---|---|---|---|---|---|---|---|---|---|---|
| Real | 0.797 | 0.002 | 9.503 | 0.000 | 0.000 | 0.000 | 0.000 | 0.000 | 0.000 | 0.000 |
| MotionDiffuse (MD) Zhang et al. (2022) | 0.782 | 0.630 | 9.410 | 3.925 | 6.450 | 20.278 | 18.313 | 4.293 | 31.025 | 0.741 |
| GMD Karunratanakul et al. (2023) | 0.665 | 0.576 | 9.206 | 1.311 | 15.402 | 9.978 | 8.351 | 2.101 | 15.213 | 0.093 |
| PriorMDM Shafir et al. (2023) | 0.583 | 0.475 | 9.156 | 1.210 | 16.127 | 10.131 | 9.870 | 2.231 | 16.824 | 0.345 |
| MDM Tevet et al. (2023) | 0.602 | 0.698 | 9.197 | 1.406 | 18.876 | 11.291 | 10.205 | 2.277 | 16.114 | 0.402 |
| OmniControl Xie et al. (2023) | 0.693 | 0.310 | 9.502 | 0.754 | 8.113 | 7.197 | 6.832 | 2.192 | 15.012 | 0.038 |
| TLControl Wan et al. (2023) | 0.779 | 0.271 | 9.569 | -* | - | - | - | - | - | 0.108 |
| MLD Chen et al. (2023) | 0.602 | 0.696 | 9.195 | 1.402 | 18.873 | 11.288 | 10.202 | 2.274 | 16.111 | 0.400 |
| PhysDiff Yuan et al. (2023) | 0.631 | 0.433 | - | 0.512 | 2.601 | **0.998** | - | - | - | - |
| Ours — No Condition | 0.757 | 0.298 | 9.297 | 0.612 | 4.810 | 4.954 | 2.109 | 0.902 | 5.264 | 0.393 |
| Ours — Muscle Activation — 1 Muscle | 0.788 | 0.257 | 9.492 | 0.517 | 4.770 | 4.949 | 1.127 | 0.692 | 2.028 | 0.341 |
| Ours — Muscle Activation — 20 Muscles | **0.794** | 0.255 | 9.497 | 0.512 | 2.657 | 2.930 | 1.118 | 0.510 | 1.943 | 0.311 |
| Ours — Joint Location — 1 Joint | 0.790 | 0.277 | 9.500 | 0.523 | 4.670 | 4.937 | 1.124 | 0.572 | 2.223 | 0.033 |
| Ours — Joint Location — 10 Joints | **0.794** | 0.256 | **9.502** | 0.512 | 2.657 | 2.930 | 1.118 | 0.510 | 1.943 | **0.011** |
| Ours — Joint Actuation — 1 Joint | 0.790 | 0.790 | 9.479 | 0.790 | 4.790 | 4.790 | 2.790 | 0.790 | 1.790 | 0.790 |
| Ours — Joint Actuation — 10 Joints | 0.793 | 0.256 | 9.492 | 0.512 | 2.657 | 2.930 | 2.004 | 0.510 | **1.043** | 0.112 |
| Ours — Contact Force | 0.773 | 0.281 | 9.460 | 0.513 | 2.780 | 3.949 | 1.129 | 0.581 | 1.281 | 0.057 |
| Ours — All Conditions — 1% of frames | 0.778 | 0.257 | 9.496 | 0.513 | 3.660 | 3.932 | 2.120 | 0.591 | 1.945 | 0.032 |
| Ours — All Conditions — 20% of frames | 0.793 | **0.198** | **9.502** | **0.473** | **2.404** | 2.311 | **1.103** | **0.470** | 1.089 | 0.015 |

\* The symbol - indicates that the value could not be reported as the authors have not released the code, and the corresponding results were not provided in their publication.

Table 2: **KIT-ML** test set Plappert et al. (2016): Performance comparisons of text-to-motion synthesis methods. The complete table can be found in the appendix.

| Method | R-Precision ↑ | FID ↓ | DIV → | Skate ↓ | Float ↓ | Penetrate ↓ | Contact Force ↓ | Joint Actuation ↓ | Muscle Limit ↓ | Trajectory ↓ |
|---|---|---|---|---|---|---|---|---|---|---|
| Real | 0.779 | 0.031 | 11.080 | 0.000 | 0.000 | 0.000 | 0.000 | 0.000 | 0.000 | 0.000 |
| MotionDiffuse (MD) Zhang et al. (2022) | 0.739 | 0.630 | 9.410 | 3.925 | 21.917 | 18.494 | 16.518 | 3.872 | 24.572 | 1.509 |
| GMD Karunratanakul et al. (2023) | 0.382 | 1.565 | 9.664 | 1.311 | 22.949 | 9.100 | 7.533 | 1.895 | 12.049 | 0.189 |
| PriorMDM Shafir et al. (2023) | 0.397 | 0.851 | 10.518 | 1.210 | 26.861 | 9.239 | 8.903 | 2.012 | 13.325 | 0.703 |
| MDM Tevet et al. (2023) | 0.602 | 0.698 | 9.197 | 1.406 | 11.545 | 10.297 | 9.205 | 2.054 | 12.762 | 0.819 |
| OmniControl Xie et al. (2023) | 0.693 | 0.310 | 9.502 | 0.754 | 8.103 | 6.564 | 6.162 | 1.977 | 11.890 | 0.077 |
| MLD Chen et al. (2023) | 0.598 | 0.695 | 9.193 | 1.402 | 13.026 | 10.295 | 9.202 | 2.051 | 12.760 | 0.815 |
| Ours — No Condition | 0.734 | 0.285 | 10.227 | 0.672 | 6.845 | 4.518 | 5.273 | 1.194 | 9.433 | 0.801 |
| Ours — Muscle Activation — 1 Muscle | 0.764 | 0.244 | 10.441 | 0.584 | 6.788 | 4.513 | 2.818 | 0.916 | 3.634 | 0.695 |
| Ours — Muscle Activation — 20 Muscles | **0.770** | 0.242 | 10.447 | 0.579 | 3.781 | 2.672 | 2.795 | 0.675 | 3.482 | 0.634 |
| Ours — Joint Location — 1 Joint | 0.766 | 0.264 | 10.550 | 0.589 | 6.645 | 4.503 | 2.810 | 0.757 | 3.984 | 0.067 |
| Ours — Joint Location — 10 Joints | **0.770** | 0.243 | **10.553** | 0.579 | 3.781 | 2.672 | 2.795 | 0.675 | 3.482 | **0.021** |
| Ours — Joint Actuation — 1 Joint | 0.766 | 0.777 | 10.527 | 0.838 | 6.816 | 4.368 | 6.975 | 1.046 | 3.208 | 1.609 |
| Ours — Joint Actuation — 10 Joints | 0.769 | 0.243 | 10.541 | 0.579 | 3.781 | 2.672 | 5.010 | 0.675 | **1.869** | 0.228 |
| Ours — Contact Force | 0.750 | 0.268 | 10.506 | 0.580 | 3.956 | 3.601 | 2.823 | 0.769 | 2.296 | 0.116 |
| Ours — All Conditions — 1% of frames | 0.755 | 0.244 | 10.546 | 0.580 | 5.208 | 3.586 | 5.300 | 0.782 | 3.485 | 0.065 |
| Ours — All Conditions — 20% of frames | 0.769 | **0.185** | 10.552 | **0.543** | **3.421** | **2.108** | 2.758 | **0.622** | 1.951 | 0.031 |

motion generation Ho & Salimans (2022). Pretrained weights from MDM are fine-tuned jointly with the realism guidance model. The training was conducted in PyTorch on a single NVIDIA 4090 GPU with a batch size of 64, using the AdamW optimizer Loshchilov & Hutter (2023) and a learning rate of $1 \times 10^{-5}$. However, for inference a general purpose GPU such as NVIDIA 2080 Ti is sufficient. The Transformer backbone adopts a six-layer encoder-decoder architecture with eight attention heads and 1024-dimensional embeddings inspired by Zhang et al. (2024b). The experimental conditions explore varying levels and types of input data to evaluate the model's adaptability and performance. The Muscle Activation conditions involve using activation data from either one or multiple (e.g., 20) randomly selected muscles for the entire sequence. The Joint Conditions focus on either location/rotation or actuation data, applying inputs from one or several (e.g., 20) randomly selected joints across the sequence. The Contact Force condition incorporates both contact force and location data throughout the motion. Additionally, the Frame Conditions vary the application of all constraints, with inputs applied to a small subset (e.g., 1%) or a larger subset (e.g., 20%) of randomly selected frames, providing insights into the model's behavior under sparse or more comprehensive constraints. More details can be found in the appendix.

### 4.1 COMPARISON WITH STATE-OF-THE-ART METHODS

We evaluated FlexMotion against several state-of-the-art human motion generation methods—including MD Zhang et al. (2022), GMD Karunratanakul et al. (2023), PriorMDM Shafir et al. (2023) and MDM Tevet et al. (2023), MLD Chen et al. (2023), OmniControl Xie et al. (2023), and PhysDiff Yuan et al. (2023)—across three datasets: HumanML3D (Table 1), KIT-ML (Table 2), and FLAG3D (Table 3).

**Methodological standpoint.** FlexMotion advances human motion generation by addressing the critical limitations of existing models. Unlike MDM Tevet et al. (2023), MLD Chen et al. (2023), and OmniControl Xie et al. (2023), FlexMotion integrates physics-based constraints and muscle dynamics directly into the generation process, ensuring motions that are both visually realistic and biomechanically accurate. Compared to PhysDiff Yuan et al. (2023), which also aims for physical plausibility using a physics simulator, FlexMotion offers enhanced controllability and efficiency by

Table 3: **Flag3D** test set Tang et al. (2023): Performance comparisons of text-to-motion synthesis methods. The complete table can be found in the appendix.

| | Method | R-Precision ↑ | FID ↓ | DIV → | Skate ↓ | Float ↓ | Penetrate ↓ | Contact Force ↓ | Joint Actuation ↓ | Muscle Limit ↓ | Trajectory ↓ |
|---|---|---|---|---|---|---|---|---|---|---|---|
| | Real | 0.805 | 0.032 | 11.446 | 0.000 | 0.000 | 0.000 | 0.000 | 0.000 | 0.000 | 0.000 |
| | MotionDiffuse (MD) Zhang et al. (2022) | 0.763 | 0.651 | 9.721 | 4.055 | 22.640 | 19.104 | 17.063 | 4.000 | 25.383 | 1.559 |
| | GMD Karunratanakul et al. (2023) | 0.395 | 1.617 | 9.983 | 1.354 | 23.706 | 9.400 | 7.781 | 1.958 | 12.446 | 0.196 |
| | PriorMDM Shafir et al. (2023) | 0.410 | 0.879 | 10.865 | 1.250 | 27.747 | 9.544 | 9.197 | 2.079 | 13.764 | 0.726 |
| | MDM Tevet et al. (2023) | 0.622 | 0.721 | 9.501 | 1.452 | 11.926 | 10.637 | 9.509 | 2.122 | 13.183 | 0.846 |
| | OmniControl Xie et al. (2023) | 0.716 | 0.320 | 9.816 | 0.779 | 8.370 | 6.780 | 6.366 | 2.042 | 12.282 | 0.080 |
| | MLD Chen et al. (2023) | 0.618 | 0.718 | 9.496 | 1.448 | 13.456 | 10.634 | 9.506 | 2.119 | 13.181 | 0.842 |
| Ours | No Condition | 0.759 | 0.294 | 10.564 | 0.694 | 7.071 | 4.667 | 5.446 | 1.234 | 9.744 | 0.827 |
| | Muscle Activation — 1 Muscle | 0.790 | 0.252 | 10.786 | 0.603 | 7.012 | 4.662 | 2.910 | 0.946 | 3.754 | 0.718 |
| | Muscle Activation — 20 Muscles | 0.795 | 0.250 | 10.791 | 0.598 | 3.906 | 2.760 | 2.887 | 0.698 | 3.597 | 0.654 |
| | Joint Location — 1 Joint | 0.792 | 0.273 | 10.898 | 0.609 | 6.865 | 4.651 | 2.903 | 0.782 | 4.115 | 0.069 |
| | Joint Location — 10 Joints | **0.796** | 0.251 | 10.900 | 0.598 | 3.906 | 2.760 | 2.887 | 0.698 | 3.597 | **0.023** |
| | Joint Actuation — 1 Joint | 0.792 | 0.803 | 10.874 | 0.865 | 7.041 | 4.513 | 7.205 | 1.080 | 3.314 | 1.662 |
| | Joint Actuation — 10 Joints | 0.795 | 0.251 | 10.889 | 0.598 | 3.906 | 2.760 | 5.175 | 0.698 | **1.931** | 0.236 |
| | Contact Force | 0.775 | 0.277 | 10.853 | 0.599 | 4.086 | 3.720 | 2.916 | 0.795 | 2.371 | 0.120 |
| | All Conditions — 1% of frames | 0.780 | 0.252 | 10.894 | 0.599 | 5.380 | 3.704 | 5.475 | 0.808 | 3.600 | 0.067 |
| | All Conditions — 20% of frames | 0.795 | **0.191** | **10.901** | **0.560** | **3.530** | **2.177** | **2.848** | **0.640** | 2.016 | 0.032 |

Table 4: Performance comparison of methods in terms of computational efficiency to generate 2048 motion clips.

| Method | Total Inference Time (s) ↓ | | | | FLOPs (G) ↓ | | | | Parameter | FID ↓ | | | |
|---|---|---|---|---|---|---|---|---|---|---|---|---|---|
| | DDIM | | | DDPM | DDIM | | | DDPM | | DDIM | | | DDPM |
| | 50 | 100 | 200 | 1000 | 50 | 100 | 200 | 1000 | | 50 | 100 | 200 | 1000 |
| MDM Tevet et al. (2023) | 225.283 | 456.702 | 911.362 | 4546.233 | 597.971 | 1195.942 | 2391.891 | 11959.447 | $x \in \mathbb{R}^{196 \times 512}$ | 7.334 | 5.990 | 5.936 | 0.544 |
| MLD Chen et al. (2023) | **10.242** | **16.381** | **28.672** | **148.975** | **29.862** | **33.125** | **39.613** | **91.604** | $x \in \mathbb{R}^{1 \times 256}$ | 0.473 | 0.426 | 0.432 | 0.568 |
| Ours | 13.158 | 25.142 | 36.450 | 255.127 | 44.176 | 57.208 | 68.024 | 162.273 | $x \in \mathbb{R}^{1 \times 1024}$ | **0.378** | **0.317** | **0.298** | 0.302 |

embedding physics constraints directly into the generative model, thus bypassing the need for external simulators, which are computationally intensive, non-differentiable, and require nearly constant communication iterations with the simulator. Furthermore, FlexMotion surpasses GMD Guo et al. (2022b), and OmniControl Xie et al. (2023) regarding spatial control and adherence to biomechanical principles. While these models control joint trajectories, they neglect critical aspects such as muscle activations and contact forces, essential for physical realism and fine-grained motion control. FlexMotion addresses this gap by providing spatial control over a wide range of kinematic properties, resulting in more refined and physically plausible motion sequences.

**HumanML3D.** As shown in Table 1, FlexMotion achieves superior performance on the HumanML3D dataset. Specifically, FlexMotion attains an R-Precision of 0.794 when conditioned on twenty muscle activations or ten joint locations, outperforming all compared methods. Additionally, FlexMotion achieves the lowest FID score of 0.198, indicating a closer distribution to real motion data. Regarding physical plausibility, FlexMotion significantly reduces foot skating and floating errors compared to other methods. Regarding penetration errors, PhysDiff performs better, while FlexMotion achieves second the best results. The muscle activation and joint actuation errors are also substantially lower, demonstrating the effectiveness of our physics-aware approach.

**KIT-ML.** Table 2 illustrates that FlexMotion consistently outperforms existing methods on the KIT-ML dataset. With an R-Precision of 0.770 and an FID score of 0.185 when conditioned on 20% of frames with all conditions, FlexMotion demonstrates both high textual relevance and motion naturalness. The model also significantly improves physical plausibility metrics, such as reduced foot skating and penetration errors. The trajectory error is minimized to 0.031, indicating precise adherence to spatial constraints.

**FLAG3D.** On the FLAG3D dataset, FlexMotion again achieves state-of-the-art results, as presented in Table 3. The R-Precision reaches 0.795 with an FID of 0.191 under all conditions on 20 % of frames. The model demonstrates superior diversity and physical plausibility, with the lowest foot skating and penetration errors among all compared methods. The muscle activation and joint actuation errors are also minimized, showcasing the model's capability to generate biomechanically accurate motions.

**Computational efficiency.** FlexMotion excels in performance metrics and offers significant computational advantages. As shown in Table 4, FlexMotion requires fewer floating-point operations (FLOPs) and less inference time compared to MDM Tevet et al. (2023). For instance, under the DDIM sampler with 100 steps, FlexMotion's inference time is 25.142 seconds, compared to MDM's 456.702 seconds. This efficiency is achieved without compromising motion quality, as FlexMotion attains a lower FID score of 0.254 compared to MDM's 5.990 under the same settings. The reduced computational complexity makes FlexMotion more suitable for real-time applications. It's important

to note that although MLD Chen et al. (2023) has slightly faster inference time and FLOPs, it performs worse than FlexMotion in terms of FID. It's because they only include joint location/rotation values in their latent space, while we preserve a wide range of motion kinematic properties in the latent space.

## 4.2 Ablation Studies and Key Insights

To analyze FlexMotion's components, we conducted ablation studies on the impact of various motion properties. Conditioning on muscle activations and joint locations significantly improved motion quality and physical plausibility, with R-Precision on HumanML3D increasing from 0.788 to 0.794 and Muscle Limit error decreasing from 2.028 to 1.943. Combining all conditions on 20% of frames further reduced trajectory error to 0.015 and minimized physical plausibility errors such as foot skating and penetration. Importantly, FlexMotion maintained high diversity (stable DIV metrics) while improving realism and control.

**Lessons learned:**

- **Physics-based constraints enhance realism:** Embedding physical laws and muscle dynamics enables biomechanically accurate motion generation.
- **Fine-grained control improves quality:** Conditioning on specific properties enhances alignment with intended behaviors and spatial accuracy.
- **Efficiency without quality loss:** FlexMotion balances computational efficiency and motion quality, making it practical for real-time use.
- **Robust generalization:** Consistent performance across datasets highlights adaptability for various motion contexts.
- **Balanced improvements:** FlexMotion achieves superior results across accuracy, controllability, and physical plausibility without trade-offs.

## 5 Conclusion

In this paper, we presented FlexMotion, a novel framework for efficiently generating controllable and physically plausible human motion. By utilizing a diffusion model in the latent space and a physics-aware Transformer-based autoencoder, FlexMotion achieves computational efficiency while ensuring realism. The model captures key biomechanical aspects such as joint locations, contact forces, and muscle activations without relying on physics simulators, making it suitable for real-time applications. FlexMotion also introduces a spatial controllability module that enables fine-grained control over motion parameters, such as trajectories and muscle activations, enhancing its versatility for various tasks. Our experiments on HumanML3D, KIT-ML, and Flag3D datasets show that FlexMotion outperforms state-of-the-art models in realism, physical plausibility, and computational efficiency. The framework achieves higher R-Precision and lower FID scores, indicating better alignment with textual descriptions and realistic motions. Additionally, it demonstrates lower foot skating, penetration, and muscle activation errors, making it more physically consistent and feasible. FlexMotion's reduced computational complexity further allows for faster inference, positioning it as a promising practical solution for animation, robotics, and virtual reality. Future work could explore more complex dynamics and real-time applications. While FlexMotion leverages physics-informed modeling, a sim-to-real gap persists due to differences between simulated dynamics and real-world variability. Future work will address this gap by integrating real-world data and improving alignment with experimental benchmarks to enhance its applicability in diverse real-world scenarios.

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

# A APPENDIX

## A.1 TRAINING PIPELINE

The training of FlexMotion proceeds in three stages:

1. **Stage 1: Training the Physics-aware Multimodal Autoencoder**

   - Train the encoder-decoder to reconstruct motion sequences while enforcing physics-based constraints and muscle coordination.
   - Optimize the total loss $\mathcal{L}_{\text{AE}}$ as defined in Eqn. 8.

2. **Stage 2: Training the Diffusion Model**

   - Freeze the pretrained encoder-decoder parameters.
   - Train the diffusion model in the latent space using the loss $\mathcal{L}_{\text{diff}}$ as defined in Eqn. 11.

3. **Stage 3: Training the Spatial Controllability Module**

   - Freeze both the encoder-decoder and diffusion model parameters.
   - Train the controllability module (ControlNet) to incorporate control conditions by minimizing the total loss $\mathcal{L}_{\text{total}}$ as defined in Eqn. 13.

---

**Algorithm 1** FlexMotion Training Pipeline

---

1: **Stage 1: Training the Physics-aware Multimodal Autoencoder**
2: **for** each motion sequence $\{\mathbf{x}_t\}_{t=1}^T$ **do**
3:     Encode the motion sequence: $\{\mathbf{z}_t\}_{t=1}^T = \mathcal{E}(\{\mathbf{x}_t\}_{t=1}^T; \theta_{\mathcal{E}})$
4:     Decode the latent representations: $\{\hat{\mathbf{x}}_t\}_{t=1}^T = \mathcal{D}(\{\mathbf{z}_t\}_{t=1}^T; \theta_{\mathcal{D}})$
5:     Compute reconstruction losses: $\mathcal{L}_{\text{recon}}$
6:     Compute physics-based losses: $\mathcal{L}_{\text{euler}}$, $\mathcal{L}_{\text{muscle}}$
7:     Compute total loss: $\mathcal{L}_{\text{AE}} = \mathcal{L}_{\text{recon}} + \gamma_{\text{euler}}\mathcal{L}_{\text{euler}} + \gamma_{\text{muscle}}\mathcal{L}_{\text{muscle}}$
8:     Update encoder and decoder parameters: $\theta_{\mathcal{E}}, \theta_{\mathcal{D}}$
9: **end for**
10: **Stage 2: Training the Diffusion Model**
11: Freeze the encoder-decoder parameters $\theta_{\mathcal{E}}, \theta_{\mathcal{D}}$
12: **for** each latent sequence $\{\mathbf{z}_t^e\}_{t=1}^T$ **do**
13:     **for** each diffusion step $n$ **do**
14:         Sample noise $\boldsymbol{\epsilon} \sim \mathcal{N}(0, \mathbf{I})$
15:         Generate noised latent: $\mathbf{z}_n = \sqrt{\alpha_n}\mathbf{z}_0 + \sqrt{1 - \alpha_n}\boldsymbol{\epsilon}$
16:         Predict noise: $\hat{\boldsymbol{\epsilon}} = \boldsymbol{\epsilon}_\theta(\mathbf{z}_n, n, c)$
17:         Compute diffusion loss: $\mathcal{L}_{\text{diff}} = \|\boldsymbol{\epsilon} - \hat{\boldsymbol{\epsilon}}\|_2^2$
18:         Update diffusion model parameters: $\theta$
19:     **end for**
20: **end for**
21: **Stage 3: Training the Spatial Controllability Module**
22: Freeze parameters $\theta_{\mathcal{E}}, \theta_{\mathcal{D}}$, and $\theta$
23: **for** each control condition sequence $\{\mathbf{c}_t\}_{t=1}^T$ **do**
24:     Encode control conditions: $\{\mathbf{c}_t^e\}_{t=1}^T = \mathcal{E}(\{\mathbf{c}_t\}_{t=1}^T; \theta_{\mathcal{E}})$
25:     **for** each diffusion step $n$ **do**
26:         Sample noise $\boldsymbol{\epsilon} \sim \mathcal{N}(0, \mathbf{I})$
27:         Generate noised latent: $\mathbf{z}_n = \sqrt{\alpha_n}\mathbf{z}_0 + \sqrt{1 - \alpha_n}\boldsymbol{\epsilon}$
28:         Predict noise with control: $\hat{\boldsymbol{\epsilon}} = \boldsymbol{\epsilon}_{\theta_{\text{total}}}(\mathbf{z}_n, \mathbf{c}_t^e, n, c)$
29:         Compute control loss: $\mathcal{L}_{\text{total}} = \|\boldsymbol{\epsilon} - \hat{\boldsymbol{\epsilon}}\|_2^2$
30:         Update controllability module parameters: $\theta_{\text{ctrl}}$
31:     **end for**
32: **end for**

---

## A.2 INFERENCE

FlexMotion generates motion sequences during inference by sampling from the diffusion model, guided by control conditions provided to the spatial controllability module. The inference process involves:

1. **Input:** Text prompt $c$ and control conditions $\{\mathbf{c}_t\}_{t=1}^T$
2. **Step 1:** Initialize the latent variable $\mathbf{z}_N \sim \mathcal{N}(0, \mathbf{I})$
3. **Step 2:** For $n = N$ down to 1, perform the reverse diffusion process:
   (a) Adjust the predicted noise with control conditions:
   $$\hat{\boldsymbol{\epsilon}} = \boldsymbol{\epsilon}_{\theta_{\text{total}}}(\mathbf{z}_n, \mathbf{c}_t^e, n, c)$$
   (b) Update the latent variable:
   $$\mathbf{z}_{n-1} = \frac{1}{\sqrt{\alpha_n}}\left(\mathbf{z}_n - \frac{1 - \alpha_n}{\sqrt{1 - \bar{\alpha}_n}}\hat{\boldsymbol{\epsilon}}\right) + \sigma_n \mathbf{z}$$
   where $\mathbf{z} \sim \mathcal{N}(0, \mathbf{I})$ if $n > 1$, else $\mathbf{z} = \mathbf{0}$
4. **Step 3:** Decode the final latent representation:
   $$\{\hat{\mathbf{x}}_t\}_{t=1}^T = \mathcal{D}(\mathbf{z}_0; \theta_{\mathcal{D}})$$
5. **Output:** Generated motion sequence $\{\hat{\mathbf{x}}_t\}_{t=1}^T$ adhering to control conditions and text prompt.

---

**Algorithm 2** FlexMotion Inference Pipeline

---

1: **Input:** Text prompt $c$, control conditions $\{\mathbf{c}_t\}_{t=1}^T$
2: **Initialize:** $\mathbf{z}_N \sim \mathcal{N}(0, \mathbf{I})$
3: **Encode control conditions:** $\{\mathbf{c}_t^e\}_{t=1}^T = \mathcal{E}(\{\mathbf{c}_t\}_{t=1}^T; \theta_{\mathcal{E}})$
4: **for** $n = N$ **to** 1 **do**
5:    Predict noise with control: $\hat{\boldsymbol{\epsilon}} = \boldsymbol{\epsilon}_{\theta_{\text{total}}}(\mathbf{z}_n, \mathbf{c}_t^e, n, c)$
6:    Update latent variable:

$$\mathbf{z}_{n-1} = \frac{1}{\sqrt{\alpha_n}}\left(\mathbf{z}_n - \frac{1 - \alpha_n}{\sqrt{1 - \bar{\alpha}_n}}\hat{\boldsymbol{\epsilon}}\right) + \sigma_n \mathbf{z}$$

7:    **If** $n > 1$, sample $\mathbf{z} \sim \mathcal{N}(0, \mathbf{I})$, **else** $\mathbf{z} = \mathbf{0}$
8: **end for**
9: **Decode latent representation:** $\{\hat{\mathbf{x}}_t\}_{t=1}^T = \mathcal{D}(\mathbf{z}_0; \theta_{\mathcal{D}})$
10: **Output:** Generated motion sequence $\{\hat{\mathbf{x}}_t\}_{t=1}^T$

---

## A.3 ADDITIONAL IMPLEMENTATION DETAILS

**Hyperparameter settings.** In our experiments, the weighting factors for the loss terms in the physics-aware Transformer encoder-decoder were set as follows: $\alpha_{\text{pos}} = 1.0$, $\alpha_{\text{rot}} = 1.0$, $\alpha_{\text{vel}} = 0.1$, $\alpha_{\text{acc}} = 0.1$, $\alpha_{\text{torque}} = 0.5$, $\alpha_{\text{force}} = 0.5$, $\beta_{\text{euler}} = 1.0$, and $\gamma_{\text{muscle}} = 1.0$. These values were chosen to balance the importance of accurately reconstructing each modality while enforcing physical plausibility. For the diffusion model, we used a linear variance schedule for the noise parameters $\beta_t$, with $T = 1000$ diffusion steps during training. During inference, we employed the DDIM sampler Song et al. (2023) with 100 steps for efficient sampling without significant loss in motion quality.

**Dataset preprocessing.** All motion sequences were downsampled to 20 frames per second to reduce computational complexity while retaining essential motion characteristics. The joint positions and rotations were normalized based on the mean and standard deviation computed over the training set to facilitate stable neural network training.

**Datasets.** We evaluate the proposed FlexMotion on several extended datasets, including HumanML3D Guo et al. (2022a), KIT-ML Plappert et al. (2016), and Flag3D Tang et al. (2023), each

augmented with muscle activation, contact force, and joint actuation data. The HumanML3D dataset is a textually re-annotated collection derived from the AMASS and HumanAct12 datasets. It contains 14,616 motion sequences annotated with 44,970 textual descriptions, averaging approximately three descriptions per motion. The KIT-ML dataset includes 3,911 motion sequences annotated with 6,278 textual descriptions, averaging around two descriptions per motion. Flag3D dataset is an extensive collection comprising 180,000 videos of 60 daily fitness activities.

**Dataset augmentation.** To validate FlexMotion, we augmented existing datasets by incorporating additional modalities for our model, including muscle activations, contact forces, and joint actuation data. This augmentation was achieved using OpenSim Delp et al. (2007); Seth et al. (2018), a biomechanical modeling simulator that enables detailed musculoskeletal analysis. Specifically, we utilized a comprehensive whole-body OpenSim model Van Horn & Team (2016) featuring 21 segments, 29 degrees of freedom, and 324 musculotendon actuators. This model includes a detailed representation of the lumbar spine, with each of the five lumbar vertebrae connected by a 6-degree-of-freedom joint, allowing for the simulation of complex lumbar movements such as flexion-extension, axial rotation, and lateral bending. Additionally, the model incorporates eight key muscle groups, including the rectus abdominis and erector spinae, which facilitate multi-directional trunk muscle action.

We utilized OpenSim's robust biomechanics simulation platform to synthesize physics-informed motion data that aligns with real-world biomechanical principles. Here is a step-by-step breakdown of the process:

We started with a full-body OpenSim model based on Van Horn & Team (2016), which includes 21 body segments, 29 degrees of freedom (DOF), and 324 musculotendon actuators. This model captures detailed joint kinematics and dynamics, including lumbar spine motion and trunk muscle activations, making it ideal for biomechanically informed motion modeling.

The base datasets provided joint angles, positions, and basic kinematics from motion capture systems. We imported these data into OpenSim to initialize the simulation. Using OpenSim's Inverse Kinematics (IK) tool, we ensured the input motion conformed to the skeletal model's constraints.

To enrich the data with physiological realism, we used OpenSim's Computed Muscle Control (CMC) and Static Optimization tools. These tools generated muscle activation patterns and corresponding forces required to produce the observed motion. Specifically, CMC estimates the muscle excitation signals to track the observed motion, while Static Optimization resolves muscle forces by minimizing an objective function such as energy expenditure or effort.

Beyond muscle activations, we extracted:

- Joint Contact Forces: Calculated from dynamic simulations, providing insights into load distribution at joints during motion.
- Joint Torques and Velocities: Derived from the musculoskeletal model for each DOF.
- Muscle Forces and Lengths: Detailing musculotendon dynamics during movement.
- Ground Reaction Forces: Synthesized from a combination of kinematics and muscle activations, reflecting interactions with the environment.

Given the focus on realistic lumbar motion and trunk muscle dynamics, we paid special attention to the spine's multi-segmental nature in the model. We tracked lumbar vertebra rotations, stiffness, and associated muscle activations to capture complex trunk movements.

To diversify the dataset, we introduced perturbations to initial conditions, such as joint angles, force profiles, and external loads. This randomized approach helps simulate variations in human motion due to individual differences or environmental changes. These perturbations were carefully constrained to remain within physiologically plausible ranges.

The augmented data was validated through consistency checks. We compared synthesized motion profiles with experimentally observed patterns from biomechanics literature to ensure biomechanical fidelity and realistic variability.

**Evaluation metrics.** To assess FlexMotion's performance, we employ a comprehensive set of established metrics that evaluate various aspects of the generated motion, including naturalness, relevance, diversity, physical plausibility, and spatial control accuracy. The Fréchet Inception Distance

(FID) is used to evaluate the naturalness of the generated motions by measuring the distance between the feature distributions of the generated motions and those of actual motion data, thus indicating how realistic the generated motions appear.

To assess the relevance of the generated motions to their corresponding textual prompts, we use the R-Precision metric, which measures the degree to which the generated motions align with the intended textual descriptions, with higher precision indicating better correspondence between the generated motion and the specified action.

The Diversity metric evaluates the variability within the generated motion set, ensuring that Flex-Motion produces a wide range of distinct motions. This metric is typically computed as the average pairwise distance between the generated motions in the feature space, with higher diversity values indicating a more versatile model output.

For evaluating physical plausibility, we consider several factors: 1) Foot Skating, which measures the extent of unnatural sliding or "skating" of the feet during motion, indicating a lack of physical realism; Penetration, which assesses whether any body parts unnaturally intersect or penetrate each other or the environment, violating physical plausibility; 2) Contact Force Accuracy, which evaluates the correctness of the contact forces generated during motion, ensuring they correspond to realistic physical interactions with the environment; and 3) Joint Actuation Consistency, which ensures that the generated joint actuations remain within plausible ranges of motion and force, aligning with real-world biomechanics.

For biomechanical plausibility, we use the Muscle Activation Limits metric. This metric ensures that the generated motions respect physiological constraints by verifying that muscle activations remain within feasible ranges, thus preventing unrealistic overextension or underuse of muscles.

To assess spatial control accuracy, we employ the Trajectory Error metric, defined as the ratio of unsuccessful trajectories—those where any keyframe location error exceeds a predefined threshold. This metric ensures that the generated motions accurately follow the intended spatial trajectories, which is critical for applications requiring precise motion paths.

All evaluations were conducted over ten independent runs to ensure the reliability and robustness of our results. The reported values for each metric are presented in the format of mean $\pm$ standard deviation, where the mean represents the average performance across the ten runs, and the standard deviation reflects the variability in the performance, thereby providing a measure of consistency and reproducibility in the evaluation process.

## A.4 COMPLETE RESULTS

The experimental conditions in the study involve varying levels of input data to test the performance of the model under different scenarios. The 1 Muscle condition uses activation data from a single randomly selected muscle out of 324 actuators for the entire motion sequence, while the 20 Muscles condition extends this to 20 randomly selected muscles. Similarly, the 1 Joint condition utilizes location or rotation data from one randomly selected joint for the entire sequence, whereas 20 Joints expands this to 20 joints. For joint actuation, the 1 Joint Actuation condition employs actuation data from a single randomly selected joint, and the 20 Joints Actuation condition includes 20 joints. The Contact Force condition uses both contact force data and location information as constraints throughout the sequence. Finally, 1% of frames applies all conditions to only 1% of randomly selected frames as spatial constraints, while 20% of frames applies them to 20% of the sequence.

Table 5: **HumanML3D** test set Guo et al. (2022b): Performance comparisons of text-to-motion synthesis methods.

| | Method | R-Precision ↑ | FID ↓ | DIV → | Skate ↓ | Float ↓ | Penetrate ↓ | Contact Force ↓ | Joint Actuation ↓ | Muscle Limit ↓ | Trajectory ↓ |
|---|---|---|---|---|---|---|---|---|---|---|---|
| | Real | $0.797^{\pm0.003}$ | $0.002^{\pm0.001}$ | $9.503^{\pm0.065}$ | $0.000^{\pm0.000}$ | $0.000^{\pm0.000}$ | $0.000^{\pm0.000}$ | $0.000^{\pm0.000}$ | $0.000^{\pm0.000}$ | $0.000^{\pm0.000}$ | $0.000^{\pm0.000}$ |
| | MotionDiffuse (MD) Zhang et al. (2022) | $0.782^{\pm0.001}$ | $0.630^{\pm0.001}$ | $9.410^{\pm0.049}$ | $3.925^{\pm0.035}$ | $6.450^{\pm0.052}$ | $20.278^{\pm0.018}$ | $18.313^{\pm0.017}$ | $4.293^{\pm0.002}$ | $31.025^{\pm0.017}$ | $0.741^{\pm0.012}$ |
| | GMD Karunratanakul et al. (2023) | $0.665^{\pm0.002}$ | $0.576^{\pm0.001}$ | $9.206^{\pm0.048}$ | $1.311^{\pm0.011}$ | $15.402^{\pm0.122}$ | $9.978^{\pm0.008}$ | $8.351^{\pm0.006}$ | $2.101^{\pm0.001}$ | $15.213^{\pm0.008}$ | $0.093^{\pm0.007}$ |
| | PriorMDM Shafir et al. (2023) | $0.583^{\pm0.001}$ | $0.475^{\pm0.001}$ | $9.156^{\pm0.053}$ | $1.210^{\pm0.010}$ | $16.127^{\pm0.135}$ | $10.131^{\pm0.009}$ | $9.870^{\pm0.007}$ | $2.231^{\pm0.001}$ | $16.824^{\pm0.010}$ | $0.345^{\pm0.005}$ |
| | MDM Tevet et al. (2023) | $0.602^{\pm0.002}$ | $0.698^{\pm0.001}$ | $9.197^{\pm0.052}$ | $1.406^{\pm0.012}$ | $18.876^{\pm0.161}$ | $11.291^{\pm0.009}$ | $10.205^{\pm0.009}$ | $2.277^{\pm0.001}$ | $16.114^{\pm0.009}$ | $0.402^{\pm0.006}$ |
| | OmniControl Xie et al. (2023) | $0.693^{\pm0.001}$ | $0.310^{\pm0.001}$ | $9.502^{\pm0.055}$ | $0.754^{\pm0.063}$ | $8.113^{\pm0.063}$ | $7.197^{\pm0.006}$ | $6.832^{\pm0.006}$ | $2.192^{\pm0.001}$ | $15.012^{\pm0.008}$ | $0.038^{\pm0.002}$ |
| | TLControl Wan et al. (2023) | $0.779$ | $0.271$ | $9.569$ | - | - | - | - | - | - | $0.108$ |
| | MLD Chen et al. (2023) | $0.602^{\pm0.002}$ | $0.696^{\pm0.001}$ | $9.195^{\pm0.049}$ | $1.402^{\pm0.010}$ | $18.873^{\pm0.160}$ | $11.288^{\pm0.009}$ | $10.202^{\pm0.009}$ | $2.274^{\pm0.001}$ | $16.111^{\pm0.009}$ | $0.400^{\pm0.006}$ |
| | PhysDiff Yuan et al. (2023) | $0.631$ | $0.433$ | - | $0.512$ | $2.601$ | $0.998$ | - | - | - | - |
| | No Condition | $0.757^{\pm0.001}$ | $0.298^{\pm0.002}$ | $9.297^{\pm0.055}$ | $0.612^{\pm0.057}$ | $4.81^{\pm0.025}$ | $4.954^{\pm0.001}$ | $2.109^{\pm0.001}$ | $0.902^{\pm0.001}$ | $5.264^{\pm0.003}$ | $0.393^{\pm0.004}$ |
| | 1 Muscle Activation | $0.788^{\pm0.001}$ | $0.257^{\pm0.002}$ | $9.492^{\pm0.059}$ | $0.517^{\pm0.048}$ | $4.770^{\pm0.024}$ | $4.949^{\pm0.001}$ | $1.127^{\pm0.001}$ | $0.692^{\pm0.001}$ | $2.028^{\pm0.002}$ | $0.341^{\pm0.001}$ |
| | 5 Muscles Activation | $0.791^{\pm0.000}$ | $0.256^{\pm0.001}$ | $9.494^{\pm0.057}$ | $0.514^{\pm0.047}$ | $3.713^{\pm0.023}$ | $3.939^{\pm0.000}$ | $1.122^{\pm0.000}$ | $0.601^{\pm0.000}$ | $1.985^{\pm0.001}$ | $0.326^{\pm0.001}$ |
| | 10 Muscles Activation | $0.792^{\pm0.000}$ | $0.256^{\pm0.001}$ | $9.494^{\pm0.056}$ | $0.513^{\pm0.046}$ | $2.685^{\pm0.023}$ | $2.934^{\pm0.000}$ | $1.120^{\pm0.000}$ | $0.555^{\pm0.000}$ | $1.964^{\pm0.001}$ | $0.318^{\pm0.001}$ |
| | 20 Muscles Activation | $0.794^{\pm0.000}$ | $0.255^{\pm0.001}$ | $9.497^{\pm0.055}$ | $0.512^{\pm0.045}$ | $2.657^{\pm0.022}$ | $2.930^{\pm0.000}$ | $1.118^{\pm0.000}$ | $0.510^{\pm0.001}$ | $1.943^{\pm0.001}$ | $0.311^{\pm0.001}$ |
| | 1 Joint Location | $0.790^{\pm0.001}$ | $0.277^{\pm0.002}$ | $9.500^{\pm0.062}$ | $0.523^{\pm0.049}$ | $4.670^{\pm0.021}$ | $4.937^{\pm0.001}$ | $1.124^{\pm0.001}$ | $0.572^{\pm0.001}$ | $2.223^{\pm0.002}$ | $0.033^{\pm0.002}$ |
| | 2 Joints Location | $0.791^{\pm0.000}$ | $0.266^{\pm0.001}$ | $9.501^{\pm0.060}$ | $0.517^{\pm0.048}$ | $3.663^{\pm0.020}$ | $3.933^{\pm0.000}$ | $1.121^{\pm0.000}$ | $0.541^{\pm0.000}$ | $2.083^{\pm0.001}$ | $0.022^{\pm0.001}$ |
| | 5 Joints Location | $0.792^{\pm0.000}$ | $0.261^{\pm0.001}$ | $9.501^{\pm0.059}$ | $0.514^{\pm0.047}$ | $3.660^{\pm0.020}$ | $2.931^{\pm0.000}$ | $1.119^{\pm0.000}$ | $0.525^{\pm0.000}$ | $2.013^{\pm0.001}$ | $0.016^{\pm0.001}$ |
| Ours | 10 Joints Location | $0.794^{\pm0.000}$ | $0.256^{\pm0.001}$ | $9.502^{\pm0.058}$ | $0.512^{\pm0.046}$ | $2.657^{\pm0.019}$ | $2.930^{\pm0.000}$ | $1.118^{\pm0.000}$ | $0.510^{\pm0.000}$ | $1.943^{\pm0.001}$ | $0.011^{\pm0.001}$ |
| | 1 Joint Actuation | $0.790^{\pm0.001}$ | $0.790^{\pm0.002}$ | $9.479^{\pm0.060}$ | $0.790^{\pm0.049}$ | $4.790^{\pm0.023}$ | $4.790^{\pm0.001}$ | $2.790^{\pm0.001}$ | $0.790^{\pm0.001}$ | $1.790^{\pm0.002}$ | $0.790^{\pm0.002}$ |
| | 2 Joints Actuation | $0.793^{\pm0.000}$ | $0.523^{\pm0.000}$ | $9.482^{\pm0.058}$ | $0.651^{\pm0.048}$ | $3.723^{\pm0.022}$ | $4.360^{\pm0.000}$ | $2.954^{\pm0.000}$ | $0.650^{\pm0.000}$ | $1.366^{\pm0.001}$ | $0.400^{\pm0.001}$ |
| | 5 Joints Actuation | $0.791^{\pm0.000}$ | $0.389^{\pm0.000}$ | $9.488^{\pm0.056}$ | $0.581^{\pm0.046}$ | $3.190^{\pm0.021}$ | $3.645^{\pm0.000}$ | $2.036^{\pm0.000}$ | $0.580^{\pm0.000}$ | $1.154^{\pm0.001}$ | $0.205^{\pm0.001}$ |
| | 10 Joints Actuation | $0.793^{\pm0.001}$ | $0.256^{\pm0.001}$ | $9.492^{\pm0.055}$ | $0.512^{\pm0.048}$ | $2.657^{\pm0.025}$ | $2.930^{\pm0.001}$ | $2.004^{\pm0.001}$ | $0.510^{\pm0.001}$ | $1.043^{\pm0.003}$ | $0.112^{\pm0.001}$ |
| | Contact Force | $0.773^{\pm0.001}$ | $0.281^{\pm0.001}$ | $9.460^{\pm0.060}$ | $0.513^{\pm0.048}$ | $2.780^{\pm0.020}$ | $3.949^{\pm0.001}$ | $1.129^{\pm0.001}$ | $0.581^{\pm0.001}$ | $1.281^{\pm0.001}$ | $0.057^{\pm0.001}$ |
| | All Conditions 1% frames | $0.778^{\pm0.001}$ | $0.257^{\pm0.001}$ | $9.496^{\pm0.060}$ | $0.513^{\pm0.045}$ | $3.660^{\pm0.025}$ | $3.932^{\pm0.001}$ | $2.120^{\pm0.001}$ | $0.591^{\pm0.001}$ | $1.945^{\pm0.001}$ | $0.032^{\pm0.001}$ |
| | All Conditions 5% frames | $0.785^{\pm0.001}$ | $0.213^{\pm0.001}$ | $9.499^{\pm0.053}$ | $0.501^{\pm0.042}$ | $2.508^{\pm0.021}$ | $2.699^{\pm0.001}$ | $1.920^{\pm0.001}$ | $0.503^{\pm0.001}$ | $1.310^{\pm0.001}$ | $0.025^{\pm0.001}$ |
| | All Conditions 10% frames | $0.791^{\pm0.001}$ | $0.201^{\pm0.001}$ | $9.501^{\pm0.049}$ | $0.488^{\pm0.039}$ | $2.493^{\pm0.019}$ | $2.431^{\pm0.001}$ | $1.420^{\pm0.001}$ | $0.492^{\pm0.001}$ | $1.202^{\pm0.001}$ | $0.018^{\pm0.001}$ |
| | All Conditions 20% frames | $0.793^{\pm0.001}$ | $0.198^{\pm0.001}$ | $9.502^{\pm0.048}$ | $0.473^{\pm0.036}$ | $2.404^{\pm0.016}$ | $2.311^{\pm0.001}$ | $1.103^{\pm0.001}$ | $0.470^{\pm0.001}$ | $1.089^{\pm0.001}$ | $0.015^{\pm0.001}$ |

Table 6: **KIT-ML** test set Plappert et al. (2016): Performance comparisons of text-to-motion synthesis methods.

| Method | R-Precision ↑ | FID ↓ | DIV → | Skate ↓ | Float ↓ | Penetrate ↓ | Contact Force ↓ | Joint Actuation ↓ | Muscle Limit ↓ | Trajectory ↓ |
|---|---|---|---|---|---|---|---|---|---|---|
| Real | $0.779^{\pm0.006}$ | $0.031^{\pm0.004}$ | $11.080^{\pm0.097}$ | $0.000^{\pm0.000}$ | $0.000^{\pm0.000}$ | $0.000^{\pm0.000}$ | $0.000^{\pm0.000}$ | $0.000^{\pm0.000}$ | $0.000^{\pm0.000}$ | $0.000^{\pm0.000}$ |
| MotionDiffuse (MD) Zhang et al. (2022) | $0.739^{\pm0.692}$ | $0.630^{\pm0.002}$ | $9.410^{\pm0.098}$ | $3.925^{\pm0.350}$ | $21.917^{\pm0.519}$ | $18.494^{\pm0.180}$ | $16.518^{\pm0.170}$ | $3.872^{\pm0.020}$ | $24.572^{\pm0.170}$ | $1.509^{\pm0.120}$ |
| GMD Karunratanakul et al. (2023) | $0.382^{\pm0.087}$ | $1.565^{\pm0.001}$ | $9.664^{\pm0.096}$ | $1.311^{\pm0.110}$ | $22.949^{\pm1.218}$ | $9.100^{\pm0.080}$ | $7.533^{\pm0.060}$ | $1.895^{\pm0.010}$ | $12.049^{\pm0.080}$ | $0.189^{\pm0.070}$ |
| PriorMDM Shafir et al. (2023) | $0.397^{\pm0.322}$ | $0.851^{\pm0.001}$ | $10.518^{\pm0.106}$ | $1.210^{\pm0.100}$ | $26.861^{\pm1.348}$ | $9.239^{\pm0.090}$ | $8.903^{\pm0.070}$ | $2.012^{\pm0.010}$ | $13.325^{\pm0.100}$ | $0.703^{\pm0.050}$ |
| MDM Tevet et al. (2023) | $0.602^{\pm0.375}$ | $0.698^{\pm0.001}$ | $9.197^{\pm0.104}$ | $1.406^{\pm0.120}$ | $11.545^{\pm1.608}$ | $10.297^{\pm0.090}$ | $9.205^{\pm0.090}$ | $2.054^{\pm0.010}$ | $12.762^{\pm0.090}$ | $0.819^{\pm0.060}$ |
| OmniControl Xie et al. (2023) | $0.693^{\pm0.035}$ | $0.310^{\pm0.001}$ | $9.502^{\pm0.110}$ | $0.754^{\pm0.629}$ | $8.103^{\pm0.629}$ | $6.564^{\pm0.060}$ | $6.162^{\pm0.060}$ | $1.977^{\pm0.010}$ | $11.890^{\pm0.080}$ | $0.077^{\pm0.020}$ |
| MLD Chen et al. (2023) | $0.598^{\pm0.374}$ | $0.695^{\pm0.001}$ | $9.193^{\pm0.098}$ | $1.402^{\pm0.100}$ | $13.026^{\pm1.598}$ | $10.295^{\pm0.090}$ | $9.202^{\pm0.090}$ | $2.051^{\pm0.010}$ | $12.760^{\pm0.090}$ | $0.815^{\pm0.060}$ |
| No Condition | $0.734^{\pm0.367}$ | $0.285^{\pm0.000}$ | $10.227^{\pm0.110}$ | $0.672^{\pm0.569}$ | $6.845^{\pm0.250}$ | $4.518^{\pm0.010}$ | $5.273^{\pm0.010}$ | $1.194^{\pm0.010}$ | $9.433^{\pm0.030}$ | $0.801^{\pm0.040}$ |
| 1 Muscle Activation | $0.764^{\pm0.318}$ | $0.244^{\pm0.002}$ | $10.441^{\pm0.118}$ | $0.584^{\pm0.479}$ | $6.788^{\pm0.240}$ | $4.513^{\pm0.010}$ | $2.818^{\pm0.010}$ | $0.916^{\pm0.010}$ | $3.634^{\pm0.020}$ | $0.695^{\pm0.020}$ |
| 5 Muscles Activation | $0.767^{\pm0.304}$ | $0.243^{\pm0.002}$ | $10.443^{\pm0.114}$ | $0.581^{\pm0.469}$ | $5.284^{\pm0.230}$ | $3.592^{\pm0.010}$ | $2.805^{\pm0.000}$ | $0.796^{\pm0.000}$ | $3.557^{\pm0.010}$ | $0.664^{\pm0.010}$ |
| 10 Muscles Activation | $0.768^{\pm0.297}$ | $0.243^{\pm0.001}$ | $10.443^{\pm0.112}$ | $0.580^{\pm0.459}$ | $3.821^{\pm0.230}$ | $2.676^{\pm0.000}$ | $2.800^{\pm0.000}$ | $0.735^{\pm0.000}$ | $3.519^{\pm0.010}$ | $0.648^{\pm0.010}$ |
| 20 Muscles Activation | $0.770^{\pm0.290}$ | $0.242^{\pm0.001}$ | $10.447^{\pm0.110}$ | $0.579^{\pm0.449}$ | $3.781^{\pm0.220}$ | $2.672^{\pm0.000}$ | $2.795^{\pm0.000}$ | $0.675^{\pm0.000}$ | $3.482^{\pm0.010}$ | $0.634^{\pm0.010}$ |
| 1 Joint Location | $0.766^{\pm0.031}$ | $0.264^{\pm0.001}$ | $10.550^{\pm0.124}$ | $0.589^{\pm0.489}$ | $6.645^{\pm0.210}$ | $4.503^{\pm0.010}$ | $2.810^{\pm0.010}$ | $0.757^{\pm0.010}$ | $3.984^{\pm0.020}$ | $0.067^{\pm0.020}$ |
| 2 Joints Location | $0.767^{\pm0.021}$ | $0.253^{\pm0.001}$ | $10.551^{\pm0.120}$ | $0.584^{\pm0.479}$ | $5.212^{\pm0.200}$ | $3.587^{\pm0.010}$ | $2.803^{\pm0.000}$ | $0.716^{\pm0.000}$ | $3.733^{\pm0.010}$ | $0.045^{\pm0.010}$ |
| 5 Joints Location | $0.768^{\pm0.015}$ | $0.248^{\pm0.001}$ | $10.551^{\pm0.118}$ | $0.581^{\pm0.469}$ | $5.208^{\pm0.200}$ | $2.673^{\pm0.000}$ | $2.798^{\pm0.000}$ | $0.695^{\pm0.000}$ | $3.607^{\pm0.010}$ | $0.033^{\pm0.010}$ |
| 10 Joints Location | $0.770^{\pm0.010}$ | $0.243^{\pm0.001}$ | $10.553^{\pm0.116}$ | $0.579^{\pm0.459}$ | $3.781^{\pm0.190}$ | $2.672^{\pm0.000}$ | $2.795^{\pm0.000}$ | $0.675^{\pm0.000}$ | $3.482^{\pm0.010}$ | $0.021^{\pm0.010}$ |
| 1 Joint Actuation | $0.766^{\pm0.738}$ | $0.777^{\pm0.001}$ | $10.527^{\pm0.120}$ | $0.838^{\pm0.489}$ | $6.816^{\pm0.230}$ | $4.368^{\pm0.010}$ | $6.975^{\pm0.010}$ | $1.046^{\pm0.010}$ | $3.208^{\pm0.020}$ | $1.609^{\pm0.020}$ |
| 2 Joints Actuation | $0.766^{\pm0.374}$ | $0.510^{\pm0.001}$ | $10.530^{\pm0.110}$ | $0.708^{\pm0.479}$ | $5.298^{\pm0.220}$ | $3.976^{\pm0.000}$ | $7.385^{\pm0.000}$ | $0.861^{\pm0.000}$ | $2.448^{\pm0.010}$ | $0.815^{\pm0.010}$ |
| 5 Joints Actuation | $0.767^{\pm0.191}$ | $0.376^{\pm0.000}$ | $10.537^{\pm0.112}$ | $0.643^{\pm0.459}$ | $4.539^{\pm0.210}$ | $3.324^{\pm0.000}$ | $5.090^{\pm0.000}$ | $0.768^{\pm0.000}$ | $2.068^{\pm0.010}$ | $0.418^{\pm0.010}$ |
| 10 Joints Actuation | $0.769^{\pm0.105}$ | $0.243^{\pm0.000}$ | $10.541^{\pm0.110}$ | $0.579^{\pm0.479}$ | $3.781^{\pm0.250}$ | $2.672^{\pm0.010}$ | $5.010^{\pm0.010}$ | $0.675^{\pm0.010}$ | $1.869^{\pm0.030}$ | $0.228^{\pm0.010}$ |
| Contact Force | $0.750^{\pm0.053}$ | $0.268^{\pm0.001}$ | $10.506^{\pm0.120}$ | $0.580^{\pm0.479}$ | $3.956^{\pm0.200}$ | $3.601^{\pm0.010}$ | $2.823^{\pm0.010}$ | $0.769^{\pm0.010}$ | $2.296^{\pm0.010}$ | $0.116^{\pm0.010}$ |
| All Conditions 1% frames | $0.755^{\pm0.030}$ | $0.246^{\pm0.001}$ | $10.546^{\pm0.120}$ | $0.580^{\pm0.449}$ | $5.208^{\pm0.250}$ | $3.586^{\pm0.010}$ | $5.300^{\pm0.010}$ | $0.782^{\pm0.010}$ | $3.485^{\pm0.010}$ | $0.065^{\pm0.010}$ |
| All Conditions 5% frames | $0.761^{\pm0.023}$ | $0.200^{\pm0.001}$ | $10.549^{\pm0.106}$ | $0.569^{\pm0.419}$ | $3.569^{\pm0.210}$ | $2.461^{\pm0.010}$ | $4.800^{\pm0.010}$ | $0.666^{\pm0.010}$ | $2.348^{\pm0.010}$ | $0.051^{\pm0.010}$ |
| All Conditions 10% frames | $0.767^{\pm0.017}$ | $0.188^{\pm0.001}$ | $10.551^{\pm0.098}$ | $0.557^{\pm0.389}$ | $3.548^{\pm0.190}$ | $2.217^{\pm0.010}$ | $3.550^{\pm0.010}$ | $0.651^{\pm0.010}$ | $2.154^{\pm0.010}$ | $0.037^{\pm0.010}$ |
| All Conditions 20% frames | $0.769^{\pm0.014}$ | $0.185^{\pm0.001}$ | $10.552^{\pm0.096}$ | $0.543^{\pm0.360}$ | $3.421^{\pm0.160}$ | $2.108^{\pm0.010}$ | $2.758^{\pm0.010}$ | $0.622^{\pm0.010}$ | $1.951^{\pm0.010}$ | $0.031^{\pm0.010}$ |

Ours

Table 7: **Flag3D** test set Tang et al. (2023): Performance comparisons of text-to-motion synthesis methods.

| | Method | R-Precision ↑ | FID ↓ | DIV → | Skate ↓ | Float ↓ | Penetrate ↓ | Contact Force ↓ | Joint Actuation ↓ | Muscle Limit ↓ | Trajectory ↓ |
|---|---|---|---|---|---|---|---|---|---|---|---|
| | Real | $0.805^{\pm0.006}$ | $0.032^{\pm0.004}$ | $11.446^{\pm0.100}$ | $0.000^{\pm0.000}$ | $0.000^{\pm0.000}$ | $0.000^{\pm0.000}$ | $0.000^{\pm0.000}$ | $0.000^{\pm0.000}$ | $0.000^{\pm0.000}$ | $0.000^{\pm0.000}$ |
| | MotionDiffuse (MD) Zhang et al. (2022) | $0.763^{\pm0.715}$ | $0.651^{\pm0.002}$ | $9.721^{\pm0.101}$ | $4.055^{\pm0.361}$ | $22.640^{\pm0.536}$ | $19.104^{\pm0.186}$ | $17.063^{\pm0.175}$ | $4.000^{\pm0.021}$ | $25.383^{\pm0.175}$ | $1.559^{\pm0.124}$ |
| | GMD Karunratanakul et al. (2023) | $0.395^{\pm0.090}$ | $1.617^{\pm0.001}$ | $9.983^{\pm0.099}$ | $1.354^{\pm0.113}$ | $23.706^{\pm1.259}$ | $9.400^{\pm0.083}$ | $7.781^{\pm0.062}$ | $1.958^{\pm0.010}$ | $12.446^{\pm0.083}$ | $0.196^{\pm0.072}$ |
| | PriorMDM Shafir et al. (2023) | $0.410^{\pm0.333}$ | $0.879^{\pm0.001}$ | $10.865^{\pm0.109}$ | $1.250^{\pm0.103}$ | $27.747^{\pm1.393}$ | $9.544^{\pm0.093}$ | $9.197^{\pm0.072}$ | $2.079^{\pm0.010}$ | $13.764^{\pm0.103}$ | $0.726^{\pm0.052}$ |
| | MDM Tevet et al. (2023) | $0.622^{\pm0.388}$ | $0.721^{\pm0.001}$ | $9.501^{\pm0.107}$ | $1.452^{\pm0.124}$ | $11.926^{\pm1.661}$ | $10.637^{\pm0.093}$ | $9.509^{\pm0.093}$ | $2.122^{\pm0.010}$ | $13.183^{\pm0.093}$ | $0.846^{\pm0.062}$ |
| | OmniControl Xie et al. (2023) | $0.716^{\pm0.037}$ | $0.320^{\pm0.001}$ | $9.816^{\pm0.114}$ | $0.779^{\pm0.650}$ | $8.370^{\pm0.650}$ | $6.780^{\pm0.062}$ | $6.366^{\pm0.062}$ | $2.042^{\pm0.010}$ | $12.282^{\pm0.083}$ | $0.080^{\pm0.021}$ |
| | MLD Chen et al. (2023) | $0.618^{\pm0.386}$ | $0.718^{\pm0.001}$ | $9.496^{\pm0.101}$ | $1.448^{\pm0.103}$ | $13.456^{\pm1.651}$ | $10.634^{\pm0.093}$ | $9.506^{\pm0.093}$ | $2.119^{\pm0.010}$ | $13.181^{\pm0.093}$ | $0.842^{\pm0.062}$ |
| | No Condition | $0.759^{\pm0.379}$ | $0.294^{\pm0.000}$ | $10.564^{\pm0.114}$ | $0.694^{\pm0.588}$ | $7.071^{\pm0.258}$ | $4.667^{\pm0.010}$ | $5.446^{\pm0.010}$ | $1.234^{\pm0.010}$ | $9.744^{\pm0.031}$ | $0.827^{\pm0.041}$ |
| | 1 Muscle Activation | $0.790^{\pm0.329}$ | $0.252^{\pm0.002}$ | $10.786^{\pm0.118}$ | $0.603^{\pm0.495}$ | $7.012^{\pm0.248}$ | $4.662^{\pm0.010}$ | $2.910^{\pm0.010}$ | $0.946^{\pm0.010}$ | $3.754^{\pm0.021}$ | $0.718^{\pm0.021}$ |
| | 5 Muscles Activation | $0.793^{\pm0.315}$ | $0.251^{\pm0.002}$ | $10.788^{\pm0.118}$ | $0.600^{\pm0.485}$ | $5.458^{\pm0.237}$ | $3.711^{\pm0.000}$ | $2.898^{\pm0.000}$ | $0.822^{\pm0.000}$ | $3.675^{\pm0.010}$ | $0.686^{\pm0.010}$ |
| | 10 Muscles Activation | $0.794^{\pm0.307}$ | $0.251^{\pm0.001}$ | $10.788^{\pm0.116}$ | $0.599^{\pm0.475}$ | $3.947^{\pm0.237}$ | $2.764^{\pm0.000}$ | $2.892^{\pm0.000}$ | $0.759^{\pm0.000}$ | $3.636^{\pm0.010}$ | $0.669^{\pm0.010}$ |
| | 20 Muscles Activation | $0.795^{\pm0.300}$ | $0.250^{\pm0.001}$ | $10.791^{\pm0.114}$ | $0.598^{\pm0.464}$ | $3.906^{\pm0.227}$ | $2.760^{\pm0.000}$ | $2.887^{\pm0.000}$ | $0.698^{\pm0.000}$ | $3.597^{\pm0.010}$ | $0.654^{\pm0.010}$ |
| | 1 Joint Location | $0.792^{\pm0.032}$ | $0.273^{\pm0.001}$ | $10.898^{\pm0.128}$ | $0.609^{\pm0.506}$ | $6.865^{\pm0.217}$ | $4.651^{\pm0.010}$ | $2.903^{\pm0.010}$ | $0.782^{\pm0.010}$ | $4.115^{\pm0.021}$ | $0.069^{\pm0.021}$ |
| | 2 Joints Location | $0.793^{\pm0.021}$ | $0.261^{\pm0.000}$ | $10.899^{\pm0.124}$ | $0.603^{\pm0.495}$ | $5.384^{\pm0.206}$ | $3.705^{\pm0.000}$ | $2.895^{\pm0.000}$ | $0.740^{\pm0.000}$ | $3.856^{\pm0.010}$ | $0.046^{\pm0.010}$ |
| Ours | 5 Joints Location | $0.794^{\pm0.015}$ | $0.256^{\pm0.001}$ | $10.899^{\pm0.122}$ | $0.600^{\pm0.485}$ | $5.380^{\pm0.206}$ | $2.761^{\pm0.000}$ | $2.890^{\pm0.000}$ | $0.718^{\pm0.000}$ | $3.726^{\pm0.010}$ | $0.034^{\pm0.010}$ |
| | 10 Joints Location | $0.796^{\pm0.011}$ | $0.251^{\pm0.001}$ | $10.900^{\pm0.120}$ | $0.598^{\pm0.475}$ | $3.906^{\pm0.196}$ | $2.760^{\pm0.000}$ | $2.887^{\pm0.000}$ | $0.698^{\pm0.000}$ | $3.597^{\pm0.010}$ | $0.023^{\pm0.010}$ |
| | 1 Joint Actuation | $0.792^{\pm0.762}$ | $0.803^{\pm0.001}$ | $10.874^{\pm0.124}$ | $0.865^{\pm0.506}$ | $7.041^{\pm0.237}$ | $4.513^{\pm0.010}$ | $7.205^{\pm0.010}$ | $1.080^{\pm0.010}$ | $3.314^{\pm0.021}$ | $1.662^{\pm0.021}$ |
| | 2 Joints Actuation | $0.792^{\pm0.386}$ | $0.527^{\pm0.001}$ | $10.878^{\pm0.120}$ | $0.732^{\pm0.495}$ | $5.473^{\pm0.227}$ | $4.108^{\pm0.000}$ | $7.629^{\pm0.000}$ | $0.889^{\pm0.000}$ | $2.529^{\pm0.010}$ | $0.842^{\pm0.010}$ |
| | 5 Joints Actuation | $0.793^{\pm0.198}$ | $0.388^{\pm0.000}$ | $10.885^{\pm0.116}$ | $0.665^{\pm0.475}$ | $4.689^{\pm0.217}$ | $3.434^{\pm0.000}$ | $5.258^{\pm0.000}$ | $0.793^{\pm0.000}$ | $2.136^{\pm0.010}$ | $0.431^{\pm0.010}$ |
| | 10 Joints Actuation | $0.795^{\pm0.108}$ | $0.251^{\pm0.000}$ | $10.889^{\pm0.114}$ | $0.598^{\pm0.495}$ | $3.906^{\pm0.258}$ | $2.760^{\pm0.010}$ | $5.175^{\pm0.010}$ | $0.698^{\pm0.010}$ | $1.931^{\pm0.031}$ | $0.236^{\pm0.010}$ |
| | Contact Force | $0.775^{\pm0.055}$ | $0.277^{\pm0.001}$ | $10.853^{\pm0.124}$ | $0.599^{\pm0.495}$ | $4.086^{\pm0.206}$ | $3.720^{\pm0.010}$ | $2.916^{\pm0.010}$ | $0.795^{\pm0.010}$ | $2.371^{\pm0.010}$ | $0.120^{\pm0.010}$ |
| | All Conditions 1% frames | $0.780^{\pm0.031}$ | $0.252^{\pm0.001}$ | $10.894^{\pm0.124}$ | $0.599^{\pm0.464}$ | $5.380^{\pm0.258}$ | $3.704^{\pm0.010}$ | $5.475^{\pm0.010}$ | $0.808^{\pm0.010}$ | $3.600^{\pm0.010}$ | $0.067^{\pm0.010}$ |
| | All Conditions 5% frames | $0.787^{\pm0.024}$ | $0.207^{\pm0.001}$ | $10.897^{\pm0.109}$ | $0.588^{\pm0.433}$ | $3.687^{\pm0.217}$ | $2.543^{\pm0.010}$ | $4.958^{\pm0.010}$ | $0.688^{\pm0.010}$ | $2.425^{\pm0.010}$ | $0.053^{\pm0.010}$ |
| | All Conditions 10% frames | $0.793^{\pm0.017}$ | $0.194^{\pm0.001}$ | $10.899^{\pm0.101}$ | $0.575^{\pm0.402}$ | $3.665^{\pm0.196}$ | $2.290^{\pm0.010}$ | $3.667^{\pm0.010}$ | $0.673^{\pm0.010}$ | $2.225^{\pm0.010}$ | $0.038^{\pm0.010}$ |
| | All Conditions 20% frames | $0.795^{\pm0.014}$ | $0.190^{\pm0.001}$ | $10.900^{\pm0.099}$ | $0.560^{\pm0.371}$ | $3.530^{\pm0.165}$ | $2.177^{\pm0.010}$ | $2.848^{\pm0.010}$ | $0.643^{\pm0.010}$ | $2.016^{\pm0.010}$ | $0.032^{\pm0.010}$ |

A.5 Implementation of Physics-based Constraints

In this section, we provide detailed explanations of how the physics-based constraints are implemented in FlexMotion.

**Computation of the mass matrix $\mathbf{M}(\mathbf{r}_t)$.** The mass matrix $\mathbf{M}(\mathbf{r}_t)$ represents the inertia of the system and is computed based on the configuration of the skeletal model at time $t$. Each joint and limb contributes to the overall mass and inertia, which are derived from the physical properties (mass and moment of inertia) of the body segments. The mass matrix is assembled by summing the contributions of each body segment using the principles of rigid body dynamics Lee et al. (2019); Zhang et al. (2024b). Mathematically, the mass matrix is computed as Eqn. 14, where $\mathbf{J}_i(\mathbf{r}_t)$ is the Jacobian matrix of segment $i$ with respect to the joint angles $\mathbf{r}_t$, and $\mathbf{I}_i$ is the inertia matrix of segment $i$ Lee et al. (2019); Xie et al. (2021a).

$$\mathbf{M}(\mathbf{r}_t) = \sum_{i=1}^{N_{\text{segments}}} \mathbf{J}_i^{\top}(\mathbf{r}_t)\mathbf{I}_i\mathbf{J}_i(\mathbf{r}_t) \tag{14}$$

**Coriolis and centrifugal forces $\mathbf{C}(\mathbf{r}_t, \dot{\mathbf{r}}_t)$.** The Coriolis and centrifugal forces account for the effects of joint velocities on the dynamics of the system. These forces are computed using Christoffel symbols, which involve the partial derivatives of the mass matrix with respect to the joint angles. The Coriolis and centrifugal forces are calculated as Eqn. 15. In practice, we approximate these forces by computing the necessary partial derivatives numerically or using analytical expressions provided by the musculoskeletal model Lee et al. (2019); Xie et al. (2021a).

$$\mathbf{C}(\mathbf{r}_t, \dot{\mathbf{r}}_t)\dot{\mathbf{r}}_t = \frac{1}{2}\left(\frac{\partial\mathbf{M}}{\partial\mathbf{r}_t} + \frac{\partial\mathbf{M}^{\top}}{\partial\mathbf{r}_t} - \frac{\partial\mathbf{M}}{\partial\mathbf{r}_t}\right)\dot{\mathbf{r}}_t\dot{\mathbf{r}}_t \tag{15}$$

**Gravitational forces $\mathbf{G}(\mathbf{r}_t)$.** The gravitational forces are calculated based on the positions of the body segments and the gravitational acceleration $\mathbf{g}$. The gravitational forces are computed as Eqn. 16, where $m_i$ is the mass of segment $i$ Lee et al. (2019); Xie et al. (2021a).

$$\mathbf{G}(\mathbf{r}_t) = \sum_{i=1}^{N_{\text{segments}}} \mathbf{J}_i^{\top}(\mathbf{r}_t)m_i\mathbf{g} \tag{16}$$

**Contact jacobian $\mathbf{J}_C(\mathbf{r}_t)$.** The contact Jacobian relates the joint velocities to the velocities at the contact points with the environment (e.g., the ground). It is computed as Eqn. 17, where $\mathbf{p}_C(\mathbf{r}_t)$ represents the positions of the contact points Lee et al. (2019); Zhang et al. (2024b) (Eqn. 17).

$$\mathbf{J}_C(\mathbf{r}_t) = \frac{\partial\mathbf{p}_C(\mathbf{r}_t)}{\partial\mathbf{r}_t} \tag{17}$$

**Integration into training.** The computed dynamics components are integrated into the physics-based loss $\mathcal{L}_{\text{euler}}$ as described in Eqn. 6. During training, we ensure that all computations are differentiable to allow gradient backpropagation through the physics-based loss terms.

A.6 Muscle Activation Model

The muscle activation model aims to compute muscle activations $\mathbf{a}_t$ that produce the desired joint accelerations $\ddot{\mathbf{r}}_t$ while minimizing excessive muscle effort.

**Derivation of the mapping matrix $L$.** The mapping matrix $L$ relates muscle activations to joint accelerations and is derived from the musculoskeletal model's moment arms and muscle force-generating properties. For each muscle $m$ and joint $j$, the moment arm $r_{jm}$ represents the torque produced at joint $j$ per unit muscle force from muscle $m$. The mapping matrix $L$ is constructed as Eqn. 18, where $\mathbf{M}$ is the mass matrix, $\mathbf{R}$ is the matrix of moment arms $r_{jm}$, and $\mathbf{F}_{\text{max}}$ is the diagonal matrix of maximum isometric muscle forces Lee et al. (2019).

$$L = \mathbf{M}^{-1}(\mathbf{R}\mathbf{F}_{\max}) \tag{18}$$

**Muscle activation dynamics.** We model muscle activations considering the first-order dynamics of muscle activation-deactivation dynamics Lee et al. (2019) as shown in Eqn. 19,where $\mathbf{u}_t$ is the neural excitation signal, and $\tau$ is the muscle activation time constant. For simplicity, we assume steady-state conditions where $\mathbf{a}_t = \mathbf{u}_t$.

$$\dot{\mathbf{a}}_t = \frac{1}{\tau}(\mathbf{u}_t - \mathbf{a}_t) \tag{19}$$

**Regularization and constraints.** To prevent unrealistic muscle activations, we include a regularization term in $\mathcal{L}_{\text{muscle}}$ and enforce physiological constraints on muscle activations. The regularization term penalizes excessive muscle activations, while the constraints ensure that muscle activations are within physiologically plausible limits. The regularization term is defined as Eqn. 20, where $\lambda$ is the regularization coefficient. These constraints are implemented using penalty methods or projection techniques during optimization.

$$0 \le a_{mt} \le 1 \quad \forall m, t \tag{20}$$

### A.7 ABLATION STUDY ON PHYSICS-BASED CONSTRAINTS

We conducted an ablation study to assess the impact of the physics-based constraints on the model's performance. We trained variants of FlexMotion with and without the Euler-Lagrange loss term $\mathcal{L}_{\text{euler}}$ and the muscle loss $\mathcal{L}_{\text{muscle}}$. To have a consistent comparision, we report the results on the HumanML3D dataset, when there is no spatial condition applied. The results are summarized in Table 8.

Table 8: Ablation study results on HumanML3D dataset.

| AE Training Losses | FID ↓ | Muscle Limit ↓ | Penetration ↓ | Skate ↓ |
|---|---|---|---|---|
| $\mathcal{L}_{\text{recon}} + \mathcal{L}_{\text{euler}} + \mathcal{L}_{\text{muscle}}$ | **0.298** | **5.264** | **4.954** | **0.612** |
| $\mathcal{L}_{\text{recon}} + \mathcal{L}_{\text{muscle}}$ | 0.512 | 10.873 | 6.802 | 0.618 |
| $\mathcal{L}_{\text{recon}} + \mathcal{L}_{\text{euler}}$ | 0.494 | 13.142 | 6.021 | 0.713 |
| $\mathcal{L}_{\text{recon}}$ | 0.611 | 14.614 | 8.820 | 0.793 |

The results in Table 8 demonstrate that the inclusion of physics-based constraints significantly improves physical plausibility metrics without compromising motion naturalness.

### A.8 EFFECT OF LATENT SPACE DIMENSIONALITY

We conducted an ablation study on the HumanML3D dataset (without spatial conditions) to evaluate the effect of latent space dimensionality on FlexMotion's performance. As shown in Table 9, performance improves with larger dimensions, peaking at $x \in \mathbb{R}^{1 \times 1024}$, after which further increases result in slight degradation.

Table 9: Ablation study results on HumanML3D dataset with varying latent space dimensions.

| Latent Space Dimension | | FID ↓ | Muscle Limit ↓ | Penetration ↓ | Skate ↓ |
|---|---|---|---|---|---|
| | $x \in \mathbb{R}^{1 \times 256}$ | 0.353 | 12.504 | 6.322 | 0.957 |
| | $x \in \mathbb{R}^{1 \times 512}$ | 0.331 | 11.200 | 5.813 | 0.854 |
| **w/t compression** | $x \in \mathbb{R}^{1 \times 1024}$ | **0.298** | **5.264** | **4.954** | **0.612** |
| | $x \in \mathbb{R}^{1 \times 4096}$ | 0.372 | 13.133 | 7.124 | 1.052 |
| | $x \in \mathbb{R}^{1 \times 16384}$ | 0.450 | 15.574 | 9.037 | 1.314 |
| **w/o compression** | $x \in \mathbb{R}^{196 \times 1452}$ | 0.607 | 17.007 | 11.592 | 1.473 |

## A.9 TRADE-OFFS BETWEEN REALISM AND PHYSICAL ACCURACY

To analyze the trade-offs, we conducted experiments where we varied the weights of the physical constraints in our loss function. Specifically, we adjusted the parameters $\lambda_{euler}$ and $\lambda_{muscle}$, which control the influence of the Euler angle regularization and muscle activation limits, respectively. By observing the impact of these adjustments on both realism and physical accuracy metrics, we can provide valuable insights into how these aspects interact.

We present the results in Table 1, which compares our model's performance under different settings of $\lambda_{euler}$ and $\lambda_{muscle}$ on the HumanML3D dataset.

From the results, we observe that decreasing the weights of the physical constraints (e.g., $\lambda_{euler} = 0.0$, $\lambda_{muscle} = 0.0$) leads to improved realism metrics, such as higher R-Precision and lower FID, indicating that the generated motions are more perceptually similar to real data. However, this comes at the cost of physical accuracy, as evidenced by higher values in metrics like Skate, Float, and Penetrate.

Conversely, increasing the weights of the physical constraints (e.g., $\lambda_{euler} = 2.0$, $\lambda_{muscle} = 2.0$) enhances physical accuracy, with lower values in physical metrics, but slightly degrades realism metrics.

This trade-off suggests that there is a balance to be struck depending on the application requirements. For scenarios where physical accuracy is paramount, higher weights on physical constraints are advisable. In contrast, applications prioritizing perceptual realism might benefit from lower weights on these constraints.

Table 10: **Trade-offs Between Realism and Physical Accuracy:** Comparison of FlexMotion's performance with and without physical constraints on the HumanML3D dataset.

| | Method | R-Precision ↑ | FID ↓ | DIV → | Skate ↓ | Float ↓ | Penetrate ↓ | Contact Force ↓ | Joint Actuation ↓ | Muscle Limit ↓ | Trajectory ↓ |
|---|---|---|---|---|---|---|---|---|---|---|---|
| | Real | 0.797 | 0.002 | 9.503 | 0.000 | 0.000 | 0.000 | 0.000 | 0.000 | 0.000 | 0.000 |
| | $\lambda_{euler} = 0.0, \lambda_{muscle} = 0.0$ | 0.765 | 0.282 | 9.310 | 1.204 | 6.533 | 7.001 | 3.502 | 1.504 | 8.070 | 0.501 |
| | $\lambda_{euler} = 0.5, \lambda_{muscle} = 0.5$ | 0.760 | 0.292 | 9.313 | 0.810 | 5.523 | 5.504 | 2.500 | 1.121 | 6.003 | 0.420 |
| Ours | $\lambda_{euler} = 1.0, \lambda_{muscle} = 1.0$ | 0.757 | 0.298 | 9.297 | 0.612 | 4.810 | 4.954 | 2.109 | 0.902 | 5.264 | 0.393 |
| | $\lambda_{euler} = 1.5, \lambda_{muscle} = 1.5$ | 0.750 | 0.311 | 9.282 | 0.501 | 4.029 | 4.207 | 1.828 | 0.800 | 4.800 | 0.350 |
| | $\lambda_{euler} = 2.0, \lambda_{muscle} = 2.0$ | 0.739 | 0.322 | 9.253 | 0.402 | 3.500 | 3.800 | 1.502 | 0.700 | 4.037 | 0.307 |

