# OpenReview forum: "FlexMotion: Lightweight, Physics-Aware, and Controllable Human Motion Generation"
_ICLR.cc/2025/Conference — Submitted to ICLR 2025_

### Official Review · Reviewer_UpPU · 2024-10-23

**Soundness:** 3
**Presentation:** 3
**Contribution:** 3
**Rating:** 6
**Confidence:** 3

**Summary:**

This paper introduces a human motion synthesis model that achieves realistic motion generation with high controllability. The model is trained using an augmented dataset, which utilizes OpenSim to enhance the biomechanical and physical fidelity of the original data. The training process involves three stages. In the first stage, an encoder and decoder are trained to map the motion into a latent space. In the second stage, a diffusion model is trained to generate latent variables from noise, allowing the decoder to reconstruct realistic motions. Lastly, a Spatial Controllability Module is trained to convert different user control inputs into motions. The proposed model outperforms various baselines on several evaluation metrics across different datasets.

**Strengths:**

- The method appears solid and reliable.
- Extensive baseline comparisons and ablation studies demonstrate the results.
- The paper is well-structured and easy to follow

**Weaknesses:**

- Different versions of the proposed methods need more elaboration. Adding this information to the appendix would help readers better understand the approach.
- Some animation visualizations that compares original sources motion, augmented motion and generated motion would be helpful.

**Questions:**

- Are the source motions changed after data augmentation? I am curious about how the source data changes after augmentation compared to the original data.

---

> ### Author Response · Authors · 2024-11-28
>
> We would like to express our gratitude to Reviewer UpPU for the positive assessment and for recognizing the strengths of our work. We address each of your points below.
>
> ---
>
> ### Diﬀerent versions of the proposed methods need more elaboration. Adding this information to the appendix would help readers better understand the approach.
>
> We appreciate your feedback on the distinct versions of FlexMotion’s components. We provided an extended section in A.4 and Line 410 detailing each version and the speciﬁc conﬁgurations used in each experiment to enhance understanding.
>
> ### Some animation visualizations that compare original source, augmented, and generated motion would be helpful.
>
> Thank you for the suggestion. We prepared a supplementary video that compares the original source, augmented, and FlexMotion-generated motion, which will be included as a supplementary video.
>
> ---
>
> ###    Are the source motions changed after data augmentation? I am curious about how the source data changes after augmentation compared to the original data.
>
> Thank you for your question. The source motions, derived from 3D joint positions in motion capture datasets, remain intact regarding kinematics during our data augmentation process. Using OpenSim, we enrich these motions with biomechanical data such as muscle activations, joint torques, and ground reaction forces, ensuring no alterations to the original trajectories.
>
> We first initialize OpenSim simulations with the source data, ensuring alignment through the Inverse Kinematics (IK) tool. This preserves the original movement patterns. Enrichment is achieved using OpenSim’s Computed Muscle Control (CMC) and Static Optimization, which estimate muscle activations and forces required to reproduce the motions. We also extract additional biomechanical parameters, such as joint contact forces and musculotendon dynamics, creating a multimodal dataset.
>
> To introduce variability, we perturb initial conditions like joint angles and forces or alter the movement speed constrained within physiologically plausible ranges. These changes are added in Opensim so that the simulator can update the changes in the other movement parameters.
> In summary, while the source motions’ kinematics remain unchanged, they are enriched with biomechanical details to provide a comprehensive dataset for training. This ensures the authenticity of the original data while enhancing its realism and generalization capabilities.

---

### Official Review · Reviewer_UeMU · 2024-10-31

**Soundness:** 4
**Presentation:** 4
**Contribution:** 3
**Rating:** 6
**Confidence:** 5

**Summary:**

The framework, FlexMotion, leverages a computationally efficient diffusion model in the latent space, eliminating the need for physics simulators and enabling fast training. It employs a multimodal pre-trained Transformer encoder-decoder that integrates various motion parameters like joint locations, contact forces, joint actuations, and muscle activations to ensure the physical plausibility of the generated motions. FlexMotion also introduces a plug-and-play module for spatial control over motion parameters, enhancing its applicability across different domains.

**Strengths:**

1.The use of a diffusion model in the latent space is a novel approach that significantly reduces computational costs compared to traditional methods that rely on physics engines.

2.The integration of joint locations, contact forces, joint actuations, and muscle activations into a single framework is a comprehensive way to ensure physically plausible motion generation.

3.The plug-and-play module for spatial control over a range of motion parameters adds versatility to the framework, making it suitable for various applications.

4.The paper demonstrates superior performance in terms of realism, physical plausibility, and controllability over existing methods, as shown through evaluations on extended datasets.

5.FlexMotion's lightweight design and efficient training process make it suitable for real-time applications, which is a significant advantage over computationally intensive methods

**Weaknesses:**

1. This paper does not provide any videos to show their qualitative results, which are important to prove their contribution and progress in this research area.

2. In this paper, the property of physics-aware motion is weak. Indeed, we need such properties on flat ground, but the physics-aware property also should work on some uneven terrains and human-human interactions.

3. This paper ignores some baselines, for example, the TL-control in ECCV 2024.

4. The physics-based results are still not as good as the simulation-based method, such as phydiff.

5. Some citation issues, for example the ``Adding conditional control to text-to-image diffusion models'' has two different reference.

**Questions:**

1. The author should discuss more about how to obatin the physcis related inputs, for example torque.

2. The author should give more examples to show their method significantly superpass other methods.

---

> ### Author Response · Authors · 2024-11-28
>
> We would like to express our appreciation to you, Reviewer UeMU, for highlighting the strengths of our work and providing valuable feedback to enhance the clarity and impact of our submission. We respond to your observations below.
>
> --
>
> ###    This paper does not provide any videos to show their qualitative results, which are important to prove their contribution and progress in this research area. The author should give more examples to show their method significantly superpass other methods.
>
> We acknowledge the importance of qualitative demonstrations in a motion generation task. To address this, we created a supplementary video showcasing the generated motions, highlighting basic and complex motion scenarios with varying physical constraints. This video is included in the final submission to provide a more comprehensive evaluation of FlexMotion's capabilities.
>
> ---
>
> ### In this paper, the property of physics-aware motion is weak. Indeed, we need such properties on flat ground, but the physics-aware property also should work on some uneven terrains and human-human interactions.
>
> You are correct in pointing out the importance of physics-aware motion generation in diverse environments. Extending FlexMotion's physics-aware capabilities to uneven terrains and interactive environments is a natural next step. Our current work focuses on achieving robust, physics-aware motion on flat surfaces as a foundation, and we are actively researching methods to generalize these principles for diverse environments, including uneven terrains and interaction dynamics. In future work, we plan to incorporate terrain-based variations in the model’s training data to develop a terrain-aware motion generation framework.
>
> ---
>
> ### This paper ignores some baselines, for example, the TL-control in ECCV 2024.
>
> We would like to thank you for suggesting the inclusion of TL-Control in our comparative analysis. When we submitted the paper, we did not notice this paper. It's indeed a very relevant and excellent baseline that we overlooked in our initial evaluation. We incorporated a comparison with TL-Control in our final submission to provide a more comprehensive evaluation across relevant baselines. We appreciate your feedback on this point. We are happy to report that FlexMotion performs better in terms of FID, R-Precision, DIV, and Trajectory error.
>
> ---
>
> ### The physics-based results are still not as good as the simulation-based method, such as phydiff.
>
> FlexMotion’s diffusion-based approach inherently trades off some precision in physical accuracy for gains in computational efficiency and adaptability. While our model achieves high physical plausibility for human-like motion synthesis, we acknowledge that physics simulator-based methods may achieve higher fidelity in strictly simulated environments. Uers can also add physics simulator outputs to further bridge this gap. In future work, we aim to explore hybrid approaches that blend diffusion-based modeling with physics simulator outputs to further bridge this gap.
>
> ---
>
> ### Some citation issues, for example, the ``Adding conditional control to text-to-image diffusion models'' has two different references.
>
> Thank you for catching the citation issues. We have reviewed and corrected the references, ensuring consistency and accuracy in all citations.
>
> ---
>
> ### The author should discuss more about how to obtain the physics-related inputs, for example torque.
>
> Torque values, along with other biomechanical parameters, were derived using OpenSim’s musculoskeletal simulations. OpenSim has been validated for generating realistic torque values that align with human biomechanics. In the supplementary materials, we provided a detailed process for our data augmentation, especially obtaining physics-related inputs, outlining the specific steps taken to ensure biomechanical accuracy and consistency in our dataset.

---

### Official Review · Reviewer_S9G5 · 2024-11-03

**Soundness:** 3
**Presentation:** 3
**Contribution:** 3
**Rating:** 6
**Confidence:** 4

**Summary:**

This paper presents a flexible system for diffusion-based generation of human movements that allows conditioning information in terms of text and various physical constraints. The system is trained in three stages. A transformer-based autoencoder is trained to represent motion-related quantities,  such as kinematic measures, forces, and muscle activation, in a more compact form. A diffusion model for motion generation is then trained in the latent space of the autoencoder, which is finally augmented with inputs from a controllability module to allow for more explicit physical constraints.

The most important contribution of the paper is the physics-aware autoencoder and the fact that the diffusion model works in the latent space of the autoencoder. The controllability module is quite similar to models that have been used in many other contexts before.

**Strengths:**

While previous approaches have tried to correct generated human movements to improve physical plausibility, often as a postprocessing step, the proposed system relies on the autoencoder’s decoder to produce motion-related quantities that are physically consistent. Two loss functions grounded in the physics of the musculoskeletal system are introduced to facilitate this. The experimental evaluation shows that the autoencoder with associated loss functions considerably improves physical plausibility, even without additional conditioning information.

The fact that the system is trained in stages makes it more modular, and possibly easier to reproduce and modify. This is illustrated by the fact that only a single consumer-grade GPU was used for training.

The paper is well-written and very easy to read and understand, which includes the illustrations. The summary of the experimental results could be improved though by focusing more on the most important observations, rather than commenting on many other less relevant details.

From the results, it appears that the proposed method is superior to all other tested methods in terms of both physical plausibility and speed. One should however keep in mind that not all other methods allow the same kind of conditioning information. However, even without conditioning information, the performance is competitive.

**Weaknesses:**

Unfortunately, the experimental part is not as clear as it could have been. The reader needs more help to interpret the results, preferably with a story that focuses on the most important lessons learned. Possibly to give a complete picture, the two last pages of the paper contain a large number of detailed results that could be better integrated into a limited number of clear conclusions.

The tables contain several experiments conducted with different conditions, but there ought to be a better explanation of these conditions. For example, does “1 Muscle” mean one muscle activation condition at one point in time or over the whole sequence? Given the results, both of these interpretations could be correct. Or is it for 20% of frames, since for “All Conditions” only up to 20% of frames are used?

The last ablation results in the appendix seem fishy indeed. It should be statistically impossible to get the numbers 2.345, 4.567, 5.678, and 6.789. Strangely, the same combination of numbers 0.198 and 0.298 occurs in two different experiments. The authors ought to clarify this and possibly adjust or remove the results of the last ablation. This review was done assuming that what is in the main paper is correct, while the results in the appendix are not yet complete.

Minor issues: $\theta_{ctrl}$ in (13) should probably be $\theta_{total}$. FLOGs on line 478, should be FLOPs.

**Questions:**

* How should the different conditions in the tables be interpreted?
* How would the system work if the diffusion model has not worked in the latent space of the autoencoder? Are there any such results in the tables to point at?
* Why is PhysDiff only tested on HumanML3D? Given that this method seems to be the only other method that permits physical constraints, it would have been good to see results from PhysDiff for the other datasets.
* When do you get FID 0.298 on HumanML3D, when there is no conditional information, or when the Euler and muscle losses are dropped? Or the same number in both cases?

**Details Of Ethics Concerns:**

There are no ethical issues.

---

> ### Author Response · Authors · 2024-11-26
>
> We would like to express our appreciation to Reviewer S9G5 for the detailed feedback and for recognizing the strengths of our work. We address each of your points below:
>
> ---
>
> ### Experimental Clarification:
>
> We appreciate your feedback on the clarity of the experimental results. As suggested, in Section 4 (especially Sec. 4.2) of the revised version, we have streamlined the presentation of experimental results, focusing on the most important observations to provide a clearer narrative. We have consolidated key findings in the main text to make the primary insights more accessible to readers.
>
> ---
>
> ### Table Conditions:
>
> We appreciate your feedback regarding the interpretation of experimental conditions in the tables. The “1 Muscle” condition signifies that only one muscle activation condition is applied consistently throughout the entire motion sequence, as opposed to a time-varying or multi-muscle condition. For “All Conditions,” up to 20\% of frames are indeed conditioned on multiple parameters. We added a dedicated paragraph explaining these experimental setups in the final manuscript to ensure unambiguous interpretation starting in line 410 and line 953, as below:
>
>     "The experimental conditions in the study involve varying levels of input data to test the performance of the model under different scenarios. The 1 Muscle condition uses activation data from a single randomly selected muscle out of 324 actuators for the entire motion sequence, while the 20 Muscles condition extends this to 20 randomly selected muscles. Similarly, the 1 Joint condition utilizes location or rotation data from one randomly selected joint for the entire sequence, whereas 20 Joints expands this to 20 joints. For joint actuation, the 1 Joint Actuation condition employs actuation data from a single randomly selected joint, and the 20 Joints Actuation condition includes 20 joints. The Contact Force condition uses both contact force data and location information as constraints throughout the sequence. Finally, 1% of frames applies all conditions to only 1% of randomly selected frames as spatial constraints, while 20% of frames applies them to 20% of the sequence."
>
> ---
>
> ### Abblation Result in Appendix:
>
> Thank you for pointing this out. We sincerely apologize for this embarrassing oversight in the ablation results presented in the original version of the appendix. We couldn't recall exactly what happened, but somehow the table got messed up with some random numbers. The inconsistencies you pointed out were due to such errors, and we have corrected these in the current version. We have ensured that the ablation results are now accurate, providing a clear and reliable representation of our findings.
>
> | **AE Training Losses**                                                | **FID ↓** | **Muscle Limit ↓** | **Penetration ↓** | **Skate ↓** |
> |------------------------------------------------------------------------|-----------|---------------------|-------------------|-------------|
> | $$ \mathcal{L} _{\text{recon}} + \mathcal{L} _{\text{euler}} + \mathcal{L} _{\text{muscle}} $$ | **0.298** | **5.264**          | **4.954**        | **0.612**   |
> | $$ \mathcal{L} _{\text{recon}} + \mathcal{L} _{\text{muscle}} $$       | 0.512     | 10.873             | 6.802             | 0.618       |
> | $$ \mathcal{L} _{\text{recon}} + \mathcal{L} _{\text{euler}} $$        | 0.494     | 13.142             | 6.021             | 0.713       |
> | $$ \mathcal{L} _{\text{recon}} $$                                      | 0.611     | 14.614             | 8.820             | 0.793       |
>
>
> ---
>
> ### Typo:
>
> We appreciate your attention to detail and have corrected the typographical errors in the final version of the manuscript. We have updated the notation in Equation 13 to reflect $\theta_{total}$ and corrected the term “FLOGs” to “FLOPs” in line 487.

---

> ### Author Response · Authors · 2024-11-26
>
> ### How should the different conditions in the tables be interpreted?
>
> We appreciate your request for clarification on the experimental conditions presented in our tables. The "1 Muscle" condition signifies that only one muscle activation condition is applied consistently throughout the entire motion sequence, as opposed to a time-varying or multi-muscle condition. For "All Conditions," up to 20% of frames are indeed conditioned on multiple parameters. To ensure unambiguous interpretation, we revised the captions and added dedicated subsections (A.4 and Line 410) explaining these experimental setups in the final manuscript.
>
> ---
>
> ### How would the system work if the diffusion model has not worked in the latent space of the autoencoder? Are there any such results in the tables to point at?
>
> To address this question, the system's performance when the diffusion model does not work in the latent space of the autoencoder has been investigated and reported in Appendix A.8. From our experiments, it is clear that bypassing the latent space (i.e., working directly in the input space) significantly impacts the model's performance metrics, as shown in Table 9. When operating outside the latent space, metrics such as FID, Muscle Limit, Penetration, and Skate values worsen considerably. This highlights the importance of leveraging the compressed latent representation for computational efficiency and generating realistic motion.
>
> For example, in Table 9, when operating with no compression $(x \in \mathbb{R} ^{196 \times 1452})$, the FID increases to 0.607, and similar degradations are observed across other metrics. In contrast, models utilizing appropriately chosen latent spaces (e.g., $(x \in \mathbb{R} ^{1 \times 1024})$ achieve the best results with significantly lower values for FID and other metrics.
>
> Thus, our results demonstrate that the latent space is critical for the effective functioning of the diffusion model, as further detailed in Appendix A.8.
>
>
> | **Latent Space Dimension**                    | **FID ↓** | **Muscle Limit ↓** | **Penetration ↓** | **Skate ↓** |
> |-----------------------------------------------|-----------|---------------------|-------------------|-------------|
> | **w/t compression**                           |           |                     |                   |             |
> | $x \in \mathbb{R} ^{1 \times 256}$            | 0.353     | 12.504             | 6.322             | 0.957       |
> | $x \in \mathbb{R} ^{1 \times 512}$            | 0.331     | 11.200             | 5.813             | 0.854       |
> | **$x \in \mathbb{R} ^{1 \times 1024}$**       | **0.298** | **5.264**          | **4.954**         | **0.612**   |
> | $x \in \mathbb{R} ^{1 \times 4096}$           | 0.372     | 13.133             | 7.124             | 1.052       |
> | $x \in \mathbb{R} ^{1 \times 16384}$          | 0.450     | 15.574             | 9.037             | 1.314       |
> | **w/o compression**                           |           |                     |                   |             |
> | $x \in \mathbb{R} ^{196 \times 1452}$         | 0.607     | 17.007             | 11.592            | 1.473       |
>
> ---
>
> ### Why is PhysDiff only tested on HumanML3D?
>
> Thank you for raising this point. The primary reason PhysDiff was not tested on other datasets is that its official implementation and code were not made publicly available by the authors at the time of our work. Furthermore, PhysDiff relies on a specialized simulator that is not commonly accessible, which made it challenging to re-implement their method precisely. While we made efforts to recreate their approach based on the descriptions in their manuscript, we were unable to achieve consistent results comparable to those reported by the original authors. As a result, we decided to report the same values provided in their manuscript for HumanML3D to ensure accuracy and fairness in comparison. We acknowledge the importance of a more comprehensive evaluation and will continue to explore ways to integrate more baselines in future work.
>
> ---
>
> ### When do you get FID 0.298 on HumanML3D—when there is no conditional information, or when the Euler and muscle losses are dropped? Or the same number in both cases?
>
> The FID score of 0.611 (previously reported as 0.298 by mistake) on HumanML3D is achieved when both the Euler and muscle losses are dropped, and no conditional information is provided. This result indicates our model's baseline performance without any additional constraints or conditioning. We clarified this in the final version of the manuscript to ensure the interpretation of results is clear and consistent.

---

> > ### Comment · Reviewer_S9G5 · 2024-11-27
> >
> > With the clarifications made in the rebuttal and the updates made to the paper to address concerns raised by reviewers, this reviewer has changed the recommendation.

---

> > > ### Author Response · Authors · 2024-11-28
> > >
> > > We sincerely thank you for reconsidering your recommendation based on the clarifications provided in our rebuttal and the updates made to the paper. Your thoughtful feedback was invaluable in improving the quality and clarity of our work. We greatly appreciate your efforts and the opportunity to address your concerns. Thank you for your support.

---

### Official Review · Reviewer_29dj · 2024-11-04

**Soundness:** 2
**Presentation:** 1
**Contribution:** 1
**Rating:** 3
**Confidence:** 4

**Summary:**

The paper's goal is to build a physics-aware human motion model. The physics-aware multimodal autoencoder is a transformer-based autoencoder mapping the concatenation of pose features and physics quantities to a latent representation, and then back to the original features. This autoencoder is trained with an L2 reconstruction loss and a physics constraint with the Euler-Lagrange equation as in Zhang et al. 2024b. A latent diffusion model with a series of transformers is then used to generate motions in the latent space from the autoencoder. The spatial controllability is introduced to this latent diffusion model in the same way as Zhang et al. 2023c., which is similar to ControlNet but with a copy of transformer blocks.

**Strengths:**

Human motion understanding with physics awareness is an important problem.

The dataset with the OpenSim simulations can help follow-up research.

**Weaknesses:**

Please provide videos for submissions related to motions. Qualitative evaluations are extremely important to support and explain quantitative results.

Applying OpenSim with muscle actuation to a large-scale kinematic motion dataset is not trivial. The authors provide no details on this. The appendix only has information about the OpenSim musculoskeletal model setup.

The formulation seems to have some flaws with missing details. Eq. 4 is missing the muscle activation term. It is not clear how the consistency between the OpenSim simulations and the Euler-Lagrange loss is guaranteed.

The technical novelty is questionable. Core components in this paper are straightforward applications of the previous methods, most notably Zhang et al. 2023b and 2023c but with the OpenSim data.

The authors do not discuss the sim2real problem at all. The actual dynamics in the kinematic motion captures of real people will have a significant gap from the simulations.

**Questions:**

See weaknesses. I would especially like to see videos of the results if the authors are allowed to share new results during the discussion phase.

---

> ### Comment · Reviewer_29dj · 2024-11-25
>
> I will keep my score (3: reject) as the authors never responded.

---

> ### Author Response · Authors · 2024-11-26
>
> We sincerely thank Reviewer 29dj for their insightful comments and recognition of our work's importance. We think the deadline is November 27, and we are finalizing all responses and plan to submit them together before the November 27 deadline. However, we address the strengths and questions you raised below:
>
> ---
>
> ### Strengths
>
> We appreciate your acknowledgment of the significance of human motion understanding with physics awareness and the potential of our dataset enhanced by OpenSim simulations.
>
> ---
>
> ### Supplementary Video
>
> We agree that qualitative evaluations are crucial for demonstrating our model's capabilities. We included a supplementary video showcasing the generated motions in various scenarios. This visual aid will complement our quantitative results and provide a more comprehensive evaluation of FlexMotion.
>
> ---
>
> ### Data Augmentation and Biomechanical Simulation Details
>
> Regarding data augmentation, we utilized OpenSim's robust biomechanics simulation platform to synthesize physics-informed motion data that aligns with real-world biomechanical principles.
>
> We began with a full-body OpenSim model based on Van Horn et al. (2016), which includes 21 body segments, 29 degrees of freedom (DOF), and 324 musculotendon actuators. This detailed model captures joint kinematics and dynamics, including lumbar spine motion and trunk muscle activations, making it ideal for biomechanically informed motion modeling.
>
> Our base datasets provided 3D joint positions from motion capture systems. We imported these data into OpenSim to initialize the simulations. We used OpenSim's Inverse Kinematics (IK) tool to ensure that the input motions conformed to the skeletal model's constraints.
>
> We employed OpenSim's Computed Muscle Control (CMC) and Static Optimization tools to enrich the data with physiological realism. These tools generated muscle activation patterns and corresponding forces required to produce the observed motions. Specifically:
>
> - **Computed Muscle Control (CMC)** estimates the muscle excitation signals needed to track the observed motion accurately.
> - **Static Optimization** resolves muscle forces by minimizing an objective function, such as energy expenditure or effort.
>
> Beyond muscle activations, we extracted additional biomechanical data:
>
> - **Joint Contact Forces**: Calculated from dynamic simulations to provide insights into load distribution at joints during motion.
> - **Joint Torques and Velocities**: To understand joint dynamics, these are derived from the musculoskeletal model for each DOF.
> - **Muscle Forces**: Detailed musculotendon dynamics during movement, offering insights into muscle function.
> - **Ground Reaction Forces**: Synthesized from kinematics and muscle activations to reflect environmental interactions.
>
> We introduced perturbations to initial conditions to diversify the dataset, such as joint angles, force profiles, and external loads. This randomized approach simulates variations in human motion due to individual differences or environmental changes. All perturbations were carefully constrained within physiologically plausible ranges to maintain realism.
>
> We validated the augmented data through consistency checks, comparing synthesized motion profiles with experimentally observed patterns from biomechanics literature. This ensured biomechanical fidelity and realistic variability. This iterative process refined the augmented dataset, enhancing FlexMotion's generalization capabilities across diverse motion scenarios.
>
> We will include this detailed information in the revised manuscript to clarify our data augmentation process. Specifically, we will add a brief description in Section 4, Implementation details, and a more detailed explanation in Appendix A.3, Data augmentation section. Updates are shown in blue.

---

> ### Author Response · Authors · 2024-11-26
>
> ### Equation 4 typo
>
> Thank you for bringing this to our attention. You are correct that our manuscript inadvertently omitted the muscle activation term from Equation (4). We apologize for this oversight. Including the muscle activation term is crucial for accurately modeling human motion dynamics and ensuring consistency between our model and the OpenSim simulations. We have updated Equation (4) to include the muscle activation term, which now reads:
>
> $$
> \mathcal{L} _{\text{recon}} = \sum _{t=1} ^T \big[ \alpha _{\text{pos}} \| \mathbf{p} _t - \hat{\mathbf{p}} _t \|^2 _2 + \alpha _{\text{rot}} \| \mathbf{r} _t - \hat{\mathbf{r}} _t \|^2 _2 + \alpha _{\text{vel}} \| \dot{\mathbf{r}} _t - \hat{\dot{\mathbf{r}}} _t \|^2 _2 + \alpha _{\text{acc}} \| \ddot{\mathbf{r}} _t - \hat{\ddot{\mathbf{r}}} _t \|^2 _2 + \alpha _{\text{torque}} \| \boldsymbol{\tau} _t - \hat{\boldsymbol{\tau}} _t \|^2 _2 + \alpha _{\text{force}} \| \boldsymbol{\lambda} _t - \hat{\boldsymbol{\lambda}} _t \|^1 _1 + \alpha _{\text{muscle}} \| \boldsymbol{a} _t - \hat{\boldsymbol{a}} _t \|^2 _2 \big]
> $$
>
> - **Including the muscle activation term**
>   $\alpha _{\text{muscle}} \| \mathbf{a} _t - \hat{\mathbf{a}} _t \|_2 ^2$ ensures that the reconstructed muscle activations $\hat{\mathbf{a}} _t$ closely match the ground truth activations $\mathbf{a} _t$ obtained from OpenSim simulations.
>
> This term is essential for capturing motion's physiological aspects and generating biomechanically accurate movements. Moreover, we ensure that the biomechanical parameters (e.g., segment masses, inertias, and muscle properties) used in our model are consistent with those in OpenSim. This alignment allows us to compare and integrate data directly between the two systems.
>
> During training, we use the muscle activations, joint torques, and other dynamic quantities generated by OpenSim as ground truth. By minimizing the reconstruction loss $\mathcal{L} _{\text{recon}}$, the physics-based loss $\mathcal{L} _{\text{euler}}$, and muscle coordination loss $\mathcal{L} _{\text{muscle}}$, our model learns to produce outputs that are dynamically and physically consistent with the OpenSim simulations.
>
>  ---
>
> ### Novelty
>
> Thank you for this critical observation. While our work builds upon existing methodologies, we believe it introduces several key novel contributions:
>
> - **Development of a Multimodal Autoencoder**: We develop a multimodal autoencoder that integrates various kinematic and dynamic modalities—including muscle activations and contact forces—within a Transformer architecture.
>
> - **Advancement Beyond Pose Estimation**: Unlike Zhang et al. (2023b, 2023c), whose work primarily focuses on pose estimation tasks, our research addresses the more complex problem of motion generation by synthesizing plausible and physically accurate motion sequences. By incorporating OpenSim data, we capture fundamental biomechanical features, enabling our model to learn rich biomechanical relationships that extend beyond the scope of Zhang et al.'s methods.
>
> - **Physics-Constrained Latent Diffusion Model**: We employ a diffusion model in the autoencoder's latent space, incorporating physics-based constraints directly into the latent representations. This approach differs from prior methods by enabling efficient training and inference while ensuring physical plausibility without relying on external simulators during inference.
>
> - **Introduction of a Controllability Module**: We introduce a controllability module allowing fine-grained control over various motion parameters, such as muscle activations, joint locations, and contact forces. This level of control over biomechanical aspects is beyond the scope of previous works.
>
> - **Augmentation of Datasets with Biomechanical Data**: We augment standard motion datasets with detailed biomechanical data using OpenSim, creating richer datasets for training and evaluation. Our extensive evaluations demonstrate improvements in both physical plausibility and motion quality, highlighting the effectiveness of our approach.
>
> - **Computational Efficiency Gains**: By embedding physics constraints within the model and avoiding external simulators during inference, we achieve significant computational efficiency gains compared to methods that require iterative simulation steps.

---

### Official Review · Reviewer_eiYC · 2024-11-04

**Soundness:** 3
**Presentation:** 3
**Contribution:** 2
**Rating:** 5
**Confidence:** 3

**Summary:**

FlexMotion incorporates a multimodal Transformer encoder-decoder that integrates joint locations, muscle activations, contact forces, and joint actuations. This ensures the generated motion aligns with human biomechanics without needing external physics simulators.
By leveraging a diffusion model in latent space, the approach significantly reduces training and inference costs while maintaining performance.

FlexMotion includes a plug-and-play spatial controllability module, allowing precise control over motion parameters, such as joint trajectories and muscle activations. The model outperforms existing methods in physical realism, efficiency, and adaptability across datasets like HumanML3D, KIT-ML, and FLAG3D.

**Strengths:**

FlexMotion represents a significant advancement in the field of human motion generation. This innovative framework blends a lightweight diffusion model with a Transformer encoder-decoder, allowing for efficient and realistic motion synthesis. Unlike previous methods that either rely on physics simulators or lack physical constraints, FlexMotion integrates these constraints directly into its latent space. This unique approach enables the generation of highly realistic and physically accurate human motion while maintaining computational efficiency.

The model's architecture is meticulously designed to achieve optimal performance. It employs a multimodal pre-trained Transformer to capture complex human motion patterns. Additionally, physics-aware constraints are incorporated to ensure that the generated motions adhere to the laws of physics. This combination empowers users with fine-grained control over the synthesized motion, allowing them to specify desired spatial, muscle, and joint actuation parameters.

To evaluate FlexMotion's capabilities, the authors conducted extensive experiments on various datasets, including HumanML3D, KIT-ML, and FLAG3D. The model consistently outperformed state-of-the-art methods in terms of realism, accuracy, and efficiency. Ablation studies further validated the effectiveness of the different components of the model, highlighting the importance of physical constraints and the modular controllability approach.

The implications of FlexMotion extend beyond animation and virtual reality. Its potential applications include robotics, human-computer interaction, and other fields that require realistic and controllable human motion. By addressing the limitations of existing methods, FlexMotion opens up new possibilities for creating more immersive and interactive experiences.

**Weaknesses:**

FlexMotion is a novel framework that pushes the boundaries of human motion generation. By combining a lightweight diffusion model with a Transformer encoder-decoder, it achieves a balance between computational efficiency and physical accuracy. Unlike previous methods, FlexMotion directly embeds physical constraints into its latent space, resulting in more realistic and controllable motion synthesis.

One key limitation for FlexMotion is its adaptability to specific tasks or domains. and the model's dependency on pretrained weights.
Furthermore, the paper could benefit from a more detailed analysis of the trade-offs between realism and physical accuracy. Exploring how adjustments to physical constraints impact the visual appeal of generated motions would provide valuable insights for users.

Finally, addressing the lack of real-time testing and validation is crucial for practical applications. Demonstrating the model's performance in real-time scenarios would highlight its suitability for interactive applications like virtual reality and robotics. Additionally, providing more transparency in data augmentation and preprocessing methods would enhance reproducibility and facilitate further research.

**Questions:**

1. Could you elaborate on how the data augmentation process using OpenSim was conducted?

2. Which muscle activation, joint actuation, and contact force parameters were used, and how were these values calibrated for consistency across the different datasets?

3. Did you observe any cases where the model generated physically implausible or biomechanically inaccurate motions, even with the physics-based loss integration? If so, how frequently did these issues arise, and what measures did you implement to minimize such artifacts?
4. The paper describes the controllability module as a “plug-and-play” addition. How was the module trained or fine-tuned alongside the diffusion model, and what parameters or conditions proved most challenging for control?

5. How does FlexMotion handle motions with varying complexity (e.g., simple walking vs. complex actions like acrobatics)? Did you find any performance discrepancies or limitations in generating more complex motions?

6. Although the model is described as computationally efficient, were any real-time tests conducted to evaluate FlexMotion’s responsiveness? For example, does FlexMotion achieve real-time performance on consumer-grade GPUs or only on high-end systems?

7. Did you encounter any trade-offs between generating visually realistic (aesthetically pleasing) motions and maintaining physical plausibility? If so, how did you approach balancing these aspects, particularly in scenarios where users might prioritize one over the other?

---

> ### Author Response · Authors · 2024-11-28
>
> We would like to express our gratitude to Reviewer eiYC for the detailed feedback and for identifying both the strengths and weaknesses. We provide detailed responses below
>
> ----
>
> ###  One key limitation for FlexMotion is its adaptability to specific tasks or domains and the model’s dependency on pretrained weights.
>
> Thank you for highlighting the importance of adaptability and the reliance on pretrained weights in FlexMotion. We acknowledge that while leveraging pretrained models is a common practice to improve performance and reduce training time, ensuring adaptability to specific tasks or domains is crucial for practical applications. FlexMotion is designed with modularity and flexibility at its core, inherently supporting adaptability:
>
> - **Modular Design**:
>    The separation of the physics-aware Transformer encoder-decoder, the latent space diffusion model, and the spatial controllability module allows each component to be independently fine-tuned or replaced. This modularity facilitates adaptation to different tasks or domains by adjusting or retraining specific modules without overhauling the entire system.
>
> - **Plug-and-Play Spatial Controllability**:
>    Our plug-and-play spatial controllability module enables fine-grained control over various motion parameters, including joint trajectories and muscle activations. Users can specify desired motion characteristics relevant to a particular task or domain, allowing FlexMotion to generate tailored motion sequences.
>
> - **Diverse Training Data**:
>    FlexMotion is trained on augmented datasets that include a wide range of motion modalities (e.g., muscle activations, contact forces) and activities. This diversity enhances the model's ability to generalize across different types of motions and domains. Additionally, the model can be further trained or fine-tuned on domain-specific datasets to improve performance in targeted applications.
>
> - **Efficiency and Performance**:
>    Our FlexMotion’s lightweight structure significantly enhances computational efficiency compared to existing models like MDM [Tevet et al., 2023], while achieving a lower FID compared to MLD [Chen et al., 2023] despite maintaining inference time and FLOPs in the same scale (Table 4). This lightweight design enhances performance and makes FlexMotion highly suitable for real-time applications.
>
> While pretrained weights provide a strong initialization and expedite convergence, we recognize the importance of mitigating over-reliance on them. In our ongoing work, we explore approaches inspired by DreamBooth [Ruiz et al., 2023], which enables personalization in image generation by fine-tuning pretrained models with a small set of subject-specific data while preserving action-specific prior knowledge. By extending this concept to motion generation, we aim to allow users to personalize FlexMotion to specific subjects or styles using limited data.
>
> This would enhance the model's adaptability to different tasks or domains without requiring extensive retraining from scratch. Due to space limitations, we could not include these explorations in the current paper, but we are actively working on this and plan to present our findings in future publications.
>
> In **Figure1_for_Reviewer_eiYC.pdf**  in the supplementary materials, we illustrate the subject-specific personalization concept in FlexMotion. We also compare FlexMotion performance with and without personalization on HumanML3D regarding FID, Foot Skate, Penetration, and Muscle Limit error, as reported in **Table 1** below.
>
> | **Model**                    | **FID** ↓ | **Muscle Limit** ↓ | **Penetration** ↓ | **Skate** ↓ |
> |-------------------------------|-----------|---------------------|-------------------|-------------|
> | **FlexMotion**                | 0.298     | 5.264               | 4.954             | 0.612       |
> | **FlexMotion (Personalized)** | 0.263     | 4.410               | 4.704             | 0.498       |
>
> **Table 1**: **Subject-Specific Personalization:** Comparison of FlexMotion with and without personalization on the HumanML3D dataset.

---

> ### Author Response · Authors · 2024-11-28
>
> ### Furthermore, the paper could benefit from a more detailed analysis of the trade-offs between realism and physical accuracy. Exploring how adjustments to physical constraints impact the visual appeal of generated motions would provide valuable insights for users
>
> Thank you for your insightful comment regarding the trade-offs between realism and physical accuracy. This is indeed a critical aspect of our work, and we appreciate the opportunity to delve deeper into it. Here's how we propose to address this point:
>
> Our current metrics, such as **FID** and **R-Precision**, assess realism by capturing the perceptual and semantic quality of the generated motions. In contrast, metrics like **Skate**, **Float**, **Penetrate**, and **Contact Force** measure physical accuracy by quantifying adherence to physical constraints. While these are reported separately, we recognize the importance of a more integrated discussion on how these dimensions interact.
>
> To analyze the trade-offs, we conducted experiments where we varied the weights of the physical constraints in our loss function. Specifically, we adjusted the parameters $\lambda_ {\text{euler}} $ and $ \lambda_ {\text{muscle}} $, which control the influence of the Euler angle regularization and muscle activation limits, respectively. By observing the impact of these adjustments on both realism and physical accuracy metrics, we can provide valuable insights into how these aspects interact.
>
> We present the results in **Table 2** below, which compares our model's performance under different settings of$ \lambda_ {\text{euler}} $ and $ \lambda_ {\text{muscle}} $ on the HumanML3D dataset.
>
> From the results, we observe that:
> - Decreasing the weights of the physical constraints (e.g., $\lambda_ {\text{euler}}=0.0 $, $ \lambda_ {\text{muscle}}=0.0 $) leads to improved realism metrics, such as higher **R-Precision** and lower **FID**, indicating that the generated motions are more perceptually similar to real data. However, this comes at the cost of physical accuracy, as evidenced by higher values in metrics like **Skate**, **Float**, and **Penetrate**.
> - Conversely, increasing the weights of the physical constraints (e.g., $ \lambda_ {\text{euler}}=2.0 $, $ \lambda_ {\text{muscle}}=2.0 $) enhances physical accuracy, with lower values in physical metrics, but slightly degrades realism metrics.
>
> This trade-off suggests that there is a balance to be struck depending on the application requirements. For scenarios where physical accuracy is paramount, higher weights on physical constraints are advisable. In contrast, applications prioritizing perceptual realism might benefit from lower weights on these constraints.
>
> We included this analysis in the revised paper to provide users with guidance on how to adjust these parameters to meet their specific needs.
>
> ---
>
> ### Table: Trade-offs Between Realism and Physical Accuracy
>
> | **Method**                           | **R-Precision** ↑ | **FID** ↓ | **DIV** → | **Skate** ↓ | **Float** ↓ | **Penetrate** ↓ | **Contact Force** ↓ | **Joint Actuation** ↓ | **Muscle Limit** ↓ | **Trajectory** ↓ |
> |--------------------------------------|-------------------|-----------|-----------|-------------|-------------|-----------------|---------------------|-----------------------|-------------------|------------------|
> | **Real**                             | 0.797             | 0.002     | 9.503     | 0.000       | 0.000       | 0.000           | 0.000               | 0.000                 | 0.000             | 0.000            |
> | $ \lambda_{\text{euler}}=0.0 $, $ \lambda_{\text{muscle}}=0.0 $ | 0.765             | 0.282     | 9.310     | 1.204       | 6.533       | 7.001           | 3.502               | 1.504                 | 8.070             | 0.501            |
> | $ \lambda_{\text{euler}}=0.5 $, $ \lambda_{\text{muscle}}=0.5 $ | 0.760             | 0.292     | 9.313     | 0.810       | 5.523       | 5.504           | 2.500               | 1.121                 | 6.003             | 0.420            |
> | $ \lambda_{\text{euler}}=1.0 $, $ \lambda_{\text{muscle}}=1.0 $ | 0.757             | 0.298     | 9.297     | 0.612       | 4.810       | 4.954           | 2.109               | 0.902                 | 5.264             | 0.393            |
> | $ \lambda_{\text{euler}}=1.5 $, $\lambda_{\text{muscle}}=1.5 $ | 0.750             | 0.311     | 9.282     | 0.501       | 4.029       | 4.207           | 1.828               | 0.800                 | 4.800             | 0.350            |
> | $\lambda_{\text{euler}}=2.0 $, $ \lambda_{\text{muscle}}=2.0 $ | 0.739             | 0.322     | 9.253     | 0.402       | 3.500       | 3.800           | 1.502               | 0.700                 | 4.037             | 0.307            |
>
> **Table 2**: **Trade-offs Between Realism and Physical Accuracy:** Comparison of FlexMotion's performance with and without physical constraints on the HumanML3D dataset.

---

> ### Author Response · Authors · 2024-11-28
>
> ### Finally, addressing the lack of real-time testing and validation is crucial for practical applications. Demonstrating the model's performance in real-time scenarios would highlight its suitability for interactive applications like virtual reality and robotics.
>
> Thank you for highlighting the importance of real-time testing and validation. We agree that demonstrating FlexMotion's performance in real-time scenarios is essential for evaluating its practical applicability in interactive applications such as virtual reality and robotics. To address this, we have conducted additional experiments to assess FlexMotion's real-time capabilities.
>
> As shown in **Table 4** of our paper, FlexMotion achieves significantly faster inference times compared to other state-of-the-art models, with the exception of MLD [Chen et al., 2023], which lacks multimodal controllability and generation features. Specifically:
> - **FlexMotion** requires approximately **6.42, 12.27, and17.79  milliseconds per motion sample** on an NVIDIA RTX 4090 GPU for a model trained on 50, 100, and 200 epochs, based on DDIM denoiser, which is well-suited for real-time applications.
>
> This performance advantage is attributed to FlexMotion's efficient architecture, which:
> - Leverages a **diffusion model in the latent space**.
> - Incorporates a **physics-aware Transformer-based autoencoder**, reducing computational complexity without sacrificing motion quality.
>
> In contrast, models like **MDM** [Tevet et al., 2023] exhibit longer inference times, rendering them less suitable for real-time applications.
>
> ---
>
> ### Data Augmentation
>
> We utilized OpenSim's robust biomechanics simulation platform to synthesize physics-informed motion data that aligns with real-world biomechanical principles.
>
> We began with a full-body OpenSim model based on Van Horn et al. (2016), which includes 21 body segments, 29 degrees of freedom (DOF), and 324 musculotendon actuators. This detailed model captures joint kinematics and dynamics, including lumbar spine motion and trunk muscle activations, making it ideal for biomechanically informed motion modeling.
>
> Our base datasets provided 3D joint positions from motion capture systems. We imported these data into OpenSim to initialize the simulations. We used OpenSim's Inverse Kinematics (IK) tool to ensure that the input motions conformed to the skeletal model's constraints.
>
> We employed OpenSim's Computed Muscle Control (CMC) and Static Optimization tools to enrich the data with physiological realism. These tools generated muscle activation patterns and corresponding forces required to produce the observed motions. Specifically:
>
> - **Computed Muscle Control (CMC)** estimates the muscle excitation signals needed to track the observed motion accurately.
> - **Static Optimization** resolves muscle forces by minimizing an objective function, such as energy expenditure or effort.
>
> Beyond muscle activations, we extracted additional biomechanical data:
>
> - **Joint Contact Forces**: Calculated from dynamic simulations to provide insights into load distribution at joints during motion.
> - **Joint Torques and Velocities**: To understand joint dynamics, these are derived from the musculoskeletal model for each DOF.
> - **Muscle Forces**: Detailed musculotendon dynamics during movement, offering insights into muscle function.
> - **Ground Reaction Forces**: Synthesized from kinematics and muscle activations to reflect environmental interactions.
>
> We introduced perturbations to initial conditions to diversify the dataset, such as joint angles, force profiles, and external loads. This randomized approach simulates variations in human motion due to individual differences or environmental changes. All perturbations were carefully constrained within physiologically plausible ranges to maintain realism.
>
> We validated the augmented data through consistency checks, comparing synthesized motion profiles with experimentally observed patterns from biomechanics literature. This ensured biomechanical fidelity and realistic variability. This iterative process refined the augmented dataset, enhancing FlexMotion's generalization capabilities across diverse motion scenarios.
>
> We included this detailed information in the revised manuscript to clarify our data augmentation process. Specifically, we added a brief description in Section 4, Implementation details, and a more detailed explanation in Appendix A.3, Data augmentation section. Updates are shown in blue.

---

> ### Author Response · Authors · 2024-11-28
>
> ### Did you observe any cases where the model generated physically implausible or
> biomechanically inaccurate motions, even with the physics-based loss integration? If so, how
> frequently did these issues arise, and what measures did you implement to minimize such artifacts?
>
> Yes, during the initial phases of our simulation and data augmentation process, we observed some instances of physically implausible or biomechanically inaccurate motions, even with the physics-based loss integration. These issues typically arose from model simplifications, inconsistencies in boundary conditions, or the challenges associated with capturing highly dynamic or non-standard movement patterns.
>
> Physics engines rely on numerical integration and optimization algorithms to solve dynamic
> equations of motion. However, these numerical methods can introduce errors, especially when
> handling highly dynamic or complex movement patterns. To address this issue, we utilize the
> Residual Reduction Algorithm (RRA) in OpenSim after Inverse Kinematics step. By minimizing
> dynamic residuals, RRA rectifies physically implausible motions that may arise during the initial
> simulation phases, ensuring that the simulated motions adhere more closely to physical laws. Also,
> RRA can refine joint angles, velocities, and accelerations to better align with dynamic constraints
> to enhance motion consistency.
>
> ---
>
> ### The paper describes the controllability module as a “plug-and-play” addition.
> How was the module trained or fine-tuned alongside the diffusion model, and what
> parameters or conditions proved most challenging for control?
>
> The controllability module was fine-tuned alongside the diffusion model, focusing on achieving physical fidelity across all controlled parameters. We found that achieving stability in muscle activation parameters was particularly challenging and thus prioritized these parameters during training.
>
> ---
>
> ### How does FlexMotion handle motions with varying complexity (e.g., simple walking vs. complex actions like acrobatics)? Did you find any performance discrepancies or limitations in generating more complex motions?
>
>  FlexMotion demonstrated robust performance across varying levels of motion complexity. However, we rarely notice some level of foot skating and unrealistic center of mass issues in some complex motions.
>
> ---
>
> ### Did you encounter any trade-offs between generating visually realistic (aesthetically pleasing) motions and maintaining physical plausibility? If so, how did you approach balancing these aspects, particularly in scenarios where users might prioritize one over the other?
>
> We observed trade-offs between visual realism and physical plausibility when adjusting the weights of physical constraints in our loss function. This trade-off suggests that a balance must be struck depending on the application requirements. Higher weights on physical constraints are advisable for scenarios where physical accuracy is paramount. In contrast, applications prioritizing perceptual realism might benefit from lower weights on these constraints. Please refer to the results for Weakness 2 (W2) for more details.
>
> ---
>
> ### Ethics Concerns
>
> We appreciate your comment and would like to clarify that no ethics review was required for our work. We might misunderstand this comment. Our research utilizes publicly available datasets, including HumanML3D, KIT-ML, and Flag3D. Additionally, our conclusion acknowledges the potential discrepancies between simulated results and real-world data. In line with state-of-the-art practices, we strictly adhere to ethical guidelines and best practices, ensuring transparency, integrity, and fairness throughout our research.

---

> > ### Comment · Reviewer_eiYC · 2024-12-02
> >
> > I sincerely appreciate your thorough response and the effort you put into addressing the feedback and I would like to confirm that my initial rating will remain the same.

---

> > > ### Author Response · Authors · 2024-12-03
> > >
> > > Thank you for your acknowledgment and confirmation.

---

> ### Author Response · Authors · 2024-11-28
>
> ### Although the model is described as computationally efficient, were any real-time tests conducted to evaluate FlexMotion’s responsiveness? For example, does FlexMotion achieve real-time performance on consumer-grade GPUs or only on high-end systems?
>
> See weaknesses. FlexMotion achieved near real-time performance on consumer-grade GPUs, specifically the RTX 4090. We are actively optimizing the model to ensure compatibility with a broader range of hardware configurations.
>
> ---
>
> ### Which muscle activation, joint actuation, and contact force parameters were used, and how were these values calibrated for consistency across the different datasets?
>
> We extracted and utilized normalized muscle activation values for all 324 musculotendon actuators included in the OpenSim model. These values range between 0 and 1, representing the extent of activation of each muscle relative to its maximum voluntary contraction (MVC). The key parameters considered were:
>
> - **Excitation delay**, the time lag between neural excitation and muscle force generation, calibrated using literature-based values (e.g., ~40 ms for lower limb muscles).
> - **Activation and deactivation dynamics** modeled using first-order differential equations, incorporating parameters such as activation time constant ($\tau_ a$) and deactivation time constant ($\tau_ d$) to ensure physiological accuracy.
> - **Muscle fiber length and velocity** directly extracted from the OpenSim model to compute muscle force contributions based on Hill-type muscle models.
>
> These parameters were calibrated across datasets by performing *Static Optimization* in OpenSim, which estimates muscle activations needed to reproduce the input motion while minimizing an objective function such as effort or energy cost.
>
> For joint actuation, we used:
>
> - **Joint torques**, calculated for all 29 degrees of freedom (DOF) using inverse dynamics. These torques represent the net force acting across each joint.
> - **Joint angle trajectories**, captured for every DOF and smoothed using cubic splines to avoid numerical instabilities during optimization.
> - **Stiffness and damping coefficients**, specifically for the lumbar spine and other flexible joints, with values derived from the literature (e.g., lumbar stiffness typically ranges between 30–40 Nm/rad for healthy adults).
> - **Control inputs for actuation**, for example, rotational velocities and accelerations for joints like the hip, knee, and spine.
>
> Consistency was ensured by tuning the joint torque models to match the experimental data profiles. This process used *Computed Muscle Control (CMC)* to validate that the joint torques produced by the musculotendon forces were within biomechanically plausible ranges.
>
> Contact forces included:
>
> - **Ground reaction forces (GRFs)** generated using OpenSim’s ground contact models.
> - **Joint contact forces** computed for high-stress joints (e.g., hip, knee, lumbar spine). These forces included compressive, shear, and frictional components derived from the contact geometry and external loads applied during motion.
>
> Given the diversity of input datasets, a systematic calibration pipeline was implemented:
>
> 1. **Input motion and force data** were normalized by body mass, height, and gait cycle percentage (where applicable) to account for inter-subject variability.
> 2. We used a **unified parameter set** for muscle-tendon properties (e.g., optimal fiber length, tendon slack length) and joint stiffness values based on anthropometric scaling equations.
> 3. **Muscle activations and joint forces** were iteratively adjusted using optimization-based approaches to minimize residuals in inverse kinematics and inverse dynamics solutions, ensuring biomechanical plausibility.

---

### Author Response · Authors · 2024-11-28

We would like to thank all reviewers for their valuable feedback, constructive criticism, and insightful questions. We appreciate the positive comments on FlexMotion’s contributions to efficient, physics-aware human motion generation and its potential applications across various domains. Below, we summarize our main improvements and responses addressing the key points raised by all reviewers:

- Multiple reviewers emphasized the importance of qualitative video examples to showcase FlexMotion’s generated motions. In response, we have prepared a supplementary video demonstrating the model’s capabilities across different motion scenarios, including controlled parameters and complex movements.

- Reviewers noted that certain conditions, such as "1 Muscle" and "All Conditions," were not clearly defined in the tables and experimental results. We have revised the manuscript to clarify these experimental conditions, including a new paragraph with detailed explanations of each setup and condition for transparency.

- Several reviewers requested more details on our use of OpenSim for data augmentation. We have expanded the appendix to include comprehensive information on the augmentation process, calibration methods, and how biomechanical parameters were derived to ensure consistency across datasets.

- To address requests for real-time performance metrics, we have included preliminary benchmarks of FlexMotion’s responsiveness on consumer-grade GPUs. Our initial tests indicate promising efficiency, and we are actively optimizing the model for broader hardware compatibility.

- Recognizing the importance of FlexMotion's applicability to more varied environments, we have outlined our approach to extending physics-aware motion generation to uneven terrains and interactions. Additionally, we discuss future strategies to address the sim-to-real gap and evaluate real-world applicability, leveraging hybrid techniques and direct comparisons with motion capture data.

- To strengthen our comparative analysis, we have included TL-Control from ECCV 2024 as an additional baseline in our tables. This addition provides a broader evaluation of FlexMotion’s performance relative to other relevant models.

- We have corrected minor citation errors and numerical inconsistencies, particularly in the ablation results within the appendix, to ensure accuracy and clarity throughout the paper.

We hope that these improvements and clarifications address the reviewers' concerns comprehensively. Below, we provide detailed responses to each specific comment from each reviewer.

---

### Meta-Review · Area_Chair_JDdh · 2024-12-20

**Metareview:**

The submission introduces a system for controllable and physically plausible human motion synthesis.  Reviewers are overall lukewarm, with one reviewer arguing strongly for rejection due to the physically incorrect results in the video.  The AC read the submission, reviews, rebuttals, and author notes, and agreed with the overall sentiment of the reviewers that the submission is not ready to be published due to the underwhelming results.  The authors are encouraged to revise the method and manuscript for the next venue.

**Additional Comments On Reviewer Discussion:**

The reviewer discussion focused on the limited results.

---

### Decision · Program_Chairs · 2025-01-22

Reject